# ProtoVAR: Efficient Dataset Distillation
# via Prototype-Guided Visual Autoregressive Modeling

**Mingyu Wang** [1]   **Wei Jiang** [2]

## Abstract

Recent advances in generative distillation have shown strong potential in constructing high quality surrogate datasets within a fraction of the time required by optimization-based approaches. However, most existing generative solutions rely on diffusion models, which suffer from two limitations. (i) Indirect matching objectives. Their sequential denoising process makes it difficult to directly match representative prototypes. (ii) Target-agnostic generation. The generation process is often decoupled from the target task, causing the synthesized samples to drift from the desired distribution. Building on this insight, We propose ProtoVAR, a prototype-guided visual autoregressive framework. Instead of relying on latent space, ProtoVAR uses the coarse-to-fine next-scale prediction of Visual AutoRegressive (VAR) modeling to maintain semantic consistency during generation. By injecting multi-scale class prototypes, ProtoVAR enforces clear representativeness constraints while preserving diversity. A pool-based selector further distills the prototype-guided outputs into a compact, task-aligned surrogate dataset. Extensive experiments show that ProtoVAR achieves state-of-the-art performance with comparable or lower computational cost than diffusion-based distillation.

## 1. Introduction

Recent progress in machine learning has been driven by the use of increasingly large datasets and models. While such

[1]College of Control Science and Engineering, Zhejiang University, Hangzhou, China [2]School of Computer Science and Technology, Zhejiang University of Water Resources and Electric Power, Hangzhou, China. Correspondence to: Wei Jiang <jiang-wei_zju@zju.edu.cn>.

*Proceedings of the 43rd International Conference on Machine Learning*, Seoul, South Korea. PMLR 306, 2026. Copyright 2026 by the author(s).

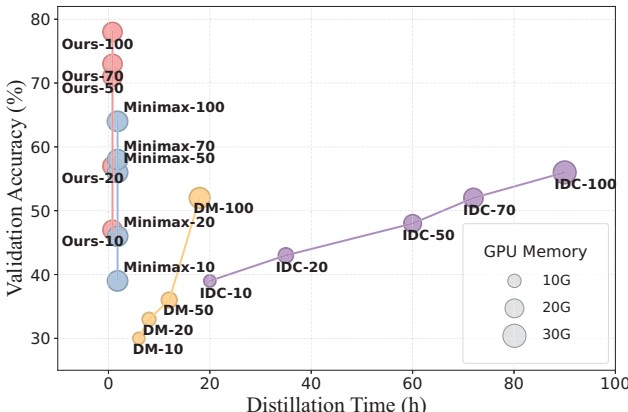

*Figure 1.* Validation accuracy and distillation time of different methods on ImageWoof. The number after each method indicates the IPC setting. Our method achieves higher efficiency and better performance compared with other approaches.

scale helps achieve strong performance, it also raises heavy demands on storage and computation, which limits their accessibility to many researchers. Since data has become the key resource that drives the progress of deep learning, this tension between performance needs and resource constraints is becoming more pronounced. To address this challenge, coreset selection (Agarwal et al., 2004) and dataset distillation (Zhao et al., 2021) have been explored as effective ways to reduce the size of training data, enabling more efficient learning under limited computational resources.

Coreset selection methods (Castro et al., 2018; Gurumoorthy et al., 2021) typically define a criterion for representativeness and then choose a set of prototypical samples from the original dataset to form a smaller one. However, since the selected samples are still drawn from the original data without creating new variations, the resulting subset is inherently limited in diversity and cannot contain information beyond the source dataset. As a result, its performance often falls below that of the full dataset. Dataset distillation (Zhao et al., 2021; Kim et al., 2022) aims to compress a large-scale dataset into a compact surrogate one, with the goal of achieving training performance comparable to that of the full dataset. Most previous approaches follow the data-matching framework, where the distilled samples are iteratively optimized so that their training effect mimics that of

of the original data (Kim et al., 2022; Zhao & Bilen, 2023; Cazenavette et al., 2022). However, as shown in Figure 1, when the Image-Per-Class (IPC) setting increases, this iterative process becomes extremely time-consuming, often exceeding the cost of training directly on the original dataset. Moreover, for fine-grained categories, the matching-based optimization struggles to provide sufficient discriminative cues, and the distilled samples frequently drift out of the original data distribution.

To tackle these challenges, recent generative dataset distillation approaches (Cazenavette et al., 2023; Gu et al., 2024a) encode dataset into the parameters of a generative model rather than compressing it into a small synthetic set. Among them, diffusion-based methods (Gu et al., 2024a; Su et al., 2024) achieve superior performance owing to their strong generative capacity. However, diffusion-based approaches still face two fundamental limitations: (i) *Indirect matching objectives.* Diffusion models generate data through sequential denoising, making direct representativeness alignment difficult. Aligning intermediate latents with the corresponding prototypes at each diffusion step is unreliable because these latents mix meaningful structure with heavy noise, weakening the guidance signal and often causing unstable learning. Therefore, most existing methods rely on indirect objectives, such as fine-tuning diffusion models or performing latent-space matching. These strategies usually optimize surrogate goals like balancing representativeness and diversity (Gu et al., 2024a). However, empirical results indicate that such surrogate optimization may not align well with the real objective due to the lack of direct control over generation, which can lead to less reliable dataset construction and limited guarantees on distribution fidelity (Santiago et al., 2025). (ii) *Target-agnostic generation.* Since diffusion-based distillation decouples the generation process from any task-relevant supervision, the synthesized data may drift away from the desired target distribution. This target ambiguity forces separate sampling for different categories and increases the risk of distributional shift. A more comprehensive review of related work is deferred to Section A.

In this work, we explore how to combine the complementary strengths of coreset selection and generative diffusion techniques to efficiently construct effective surrogate datasets. Motivated by the above analysis, we adopt Visual AutoRegressive modeling (VAR) (Tian et al., 2024) as a new generative backbone. VAR reformulates image synthesis as a coarse-to-fine next-scale prediction problem, enabling efficient and scalable visual modeling. Built upon this paradigm, we leverage coreset-identified class prototypes to guide generation across multiple spatial scales. This allows the synthesized samples to stay representative of the prototypes while gaining meaningful diversity, thus addressing the key challenges of coreset-based methods and the diffusion limitations discussed in (i). Moreover, owing to the

high efficiency of VAR (over 70× faster inference than DiT), we generate a candidate pool and use a lightweight selector to retrieve high-confidence samples. We then reapply coreset principles to obtain a compact yet representative surrogate dataset. This additional stage not only mitigates the diffusion-related limitation in (ii) but also further enhances diversity. The contributions of this work are threefold:

- We analyze the limitations inherent in both coreset selection and generative dataset distillation, and introduce a new framework that leverages the complementary strengths of these two paradigms.

- We propose a novel prototype-guided paradigm built upon VAR, which targets representative prototype matching and task-relevant synthesis, enabling more representative and discriminative distilled datasets.

- Extensive experiments show state-of-the-art performance, with +2.3% on ImageWoof and +2.5% on ImageNet over prior generative methods.

## 2. Preliminaries

### 2.1. Prototype-Based Coreset Selection

Coreset selection aims to choose a small subset of samples that can represent the structure of a large dataset. A common strategy is to construct *prototypes*, where each prototype acts as a representative point in the feature space.

Given a class dataset $\{x_i\}_{i=1}^N$ and a feature extractor $\Phi(\cdot)$, we first compute the feature mean as a reference point:

$$\bar{\Phi} = \frac{1}{N} \sum_{j=1}^N \Phi(x_j). \tag{1}$$

A single prototype can then be selected by choosing the sample whose feature is closest to this mean. This medoid-style representative is defined as:

$$p_1 = \arg\min_i \left\| \Phi(x_i) - \bar{\Phi} \right\|. \tag{2}$$

This prototype preserves semantic information while staying aligned with the original data. In later sections, we extend this idea beyond a single representative to capture broader variations within each class.

### 2.2. VAR for Distillation

To combine the strengths of coreset selection and generative distillation, we adopt VAR as the generative backbone and introduce a light-weight prototype-guided sampling pipeline tailored for dataset distillation. VAR reformulates image autoregression as a coarse-to-fine next-scale prediction problem: a latent feature map $f \in \mathbb{R}^{h \times w \times C}$ is quantized into $S$

discrete multi-scale token maps $R = (r_1, \ldots, r_S)$, where each $r_s \in [V]^{h_s \times w_s}$. The joint factorization follows:

$$P_\theta(R) = \prod_{s=1}^{S} P_\theta(r_s \mid r_{<s}), \qquad (3)$$

and during the $s$-th step all $h_s w_s$ tokens are predicted in parallel conditioned on the prefix $r_{<s}$. This next-scale design preserves spatial locality, supports efficient block-wise causal attention and KV-caching, and yields much faster inference than many diffusion backbones while maintaining strong modeling capacity.

We leverage VAR's multi-scale decoding mechanism to guide the generative process toward representative regions of the data distribution using class prototypes. Inspired by the representativeness benefits of coreset selection, we pre-compute multi-scale prototype descriptors for each class by extracting real-image features at the corresponding VAR stages. While preserving representativeness, we also exploit VAR's generation capability to introduce controlled diversity (as discussed in Section 1), ensuring that the synthesized samples remain faithful to the prototypes while capturing diverse intra-class styles. During autoregressive (AR) sampling, we construct expected continuous embeddings from the logits, decode them into a tentative latent (or image) using the VQ-VAE proxy, and compute a similarity score between the tentative output and the class prototype at the current scale. The sampler then applies a lightweight, scale-dependent correction to the per-token logits based on this similarity. Formally, the corrected logit is given by :

$$\tilde{\ell}^{(s)}(i, v) = \ell^{(s)}(i, v) + \lambda^{(s)} b^{(s)}(i, v), \qquad (4)$$

where $\ell^{(s)}(i, v)$ is the original VAR logit for token $i$ and vocabulary entry $v$ at scale $s$, $b^{(s)}(i, v)$ is a prototype-induced bias computed from the similarity between the tentative latent and the corresponding prototype (e.g., multi-scale matching scores), and $\lambda^{(s)}$ is a scale-dependent guidance weight that peaks at mid/fine scales to enhance semantic consistency and stabilize local structure generation. This formulation follows a simple principle: tokens whose expected embeddings better align with the prototype receive positive logit adjustments, providing a soft and controllable prototype-guided bias without overriding the model's inherent generative flexibility. In practice, the "expected embedding $\rightarrow$ tentative decode $\rightarrow$ prototype score $\rightarrow$ logits bias" loop is executed only at selected scales to balance guidance fidelity and computational efficiency.

## 3. Method

### 3.1. Multi-Prototype Feature Bank

Leveraging the prototype principles introduced in Section 2.1, we expand the single medoid into a compact yet diverse prototype set for each class. After obtaining the initial representative, additional prototypes are selected via a farthest-first traversal that maximizes coverage of intra-class variations. Concretely, the $k$-th prototype is chosen as:

$$p_k = \arg\max_i \; \min_{j<k} \big\| \Phi(x_i) - \Phi(x_{m_j}) \big\|, \qquad (5)$$

which ensures that each newly added prototype lies in a region of the feature space insufficiently represented by the previous ones. This greedy rule yields a small coreset that captures both semantic centrality and boundary diversity.

**Multi-Scale Feature Bank.** For each selected prototype, we further construct a set of stage-aware descriptors tailored for the VAR sampling process. Since each VAR stage corresponds to a different patch resolution, the prototype image is resized to the expected spatial scale of each stage. We then extract ResNet features to obtain a collection of multi-scale descriptors $\{p_{c,s}\}_{s=1}^{S}$, where each $p_{c,s} \in \mathbb{R}^d$ provides a scale-aligned representation of the same prototype. These descriptors collectively form a multi-scale feature bank, enabling consistent, stage-specific guidance during generation and allowing the model to anchor its denoising trajectory to semantically meaningful class exemplars across stages.

### 3.2. Prototype-Guided Logit Refinement

To incorporate prototype-level semantic control into VAR sampling, we introduce a prototype-conditioned logit refinement mechanism. The central idea is to identify which candidate tokens advance the prototype-consistent semantic direction *without* explicitly decoding all vocabulary options. This is achieved via an expected embedding relaxation together with a direction-aware logit correction rule.

**Definition 3.1** (Expected Semantic Direction)**.** Let $E \in \mathbb{R}^{V \times C}$ denote the vocabulary embedding matrix. At VAR stage $s$, the model outputs logits $\ell^{(s)} \in \mathbb{R}^{B \times L_s \times V}$, where $L_s = h_s w_s$ is the number of tokens at scale $s$. Let $\Pi^{(s)} = \mathrm{softmax}(\ell^{(s)})$. For batch index $b$ and token position $i$, we define the expected semantic embedding as

$$\bar{h}_{b,i}^{(s)} = \Pi_{b,i,:}^{(s)} E = \mathbb{E}_{v \sim \Pi_{b,i,:}^{(s)}}[E_v] \in \mathbb{R}^C, \qquad (6)$$

where $v$ indexes tokens in the vocabulary and $\Pi_{b,i,:}^{(s)}$ defines a categorical distribution over tokens. Reshaping $\{\bar{h}_{b,i}^{(s)}\}_{i=1}^{L_s}$ yields a feature map $\hat{H}_b^{(s)} \in \mathbb{R}^{C \times h_s \times w_s}$, which is decoded by the VQ-VAE proxy into a tentative image $\hat{x}_b^{(s)}$. We extract its visual feature $f(\hat{x}_b^{(s)}) \in \mathbb{R}^d$ and define the prototype similarity as

$$\mathrm{sim}_b^{(s)} = \cos\big(f(\hat{x}_b^{(s)}), \, p_{c,s}\big). \qquad (7)$$

The expected semantic embedding $\bar{h}_{b,i}^{(s)}$ serves as a differentiable surrogate for identifying which token directions are aligned with the model's current semantic trajectory.

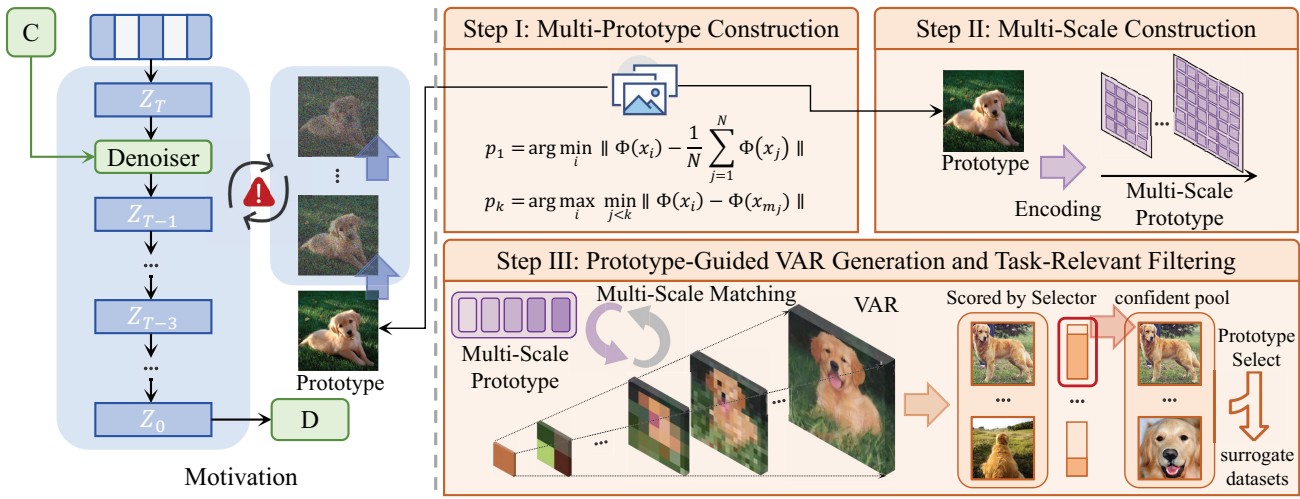

*Figure 2.* Overview of the proposed method for distilled dataset synthesis using VAR. **Left**: Most existing methods rely on latent-space matching as a surrogate objective. Since diffusion generation proceeds through sequential denoising, directly aligning prototypes with intermediate latents may unintentionally align with noise (see Section B.2). **Right**: Our method consists of prototype guidance and Target-Guided Filtering. We first construct multi-scale prototype representations aligned with VAR stage resolutions, which subsequently guide generation at corresponding scales. This produces a prototype-conditioned candidate pool, from which high-confidence samples are selected and further refined via coreset-based filtering to obtain a compact and representative distilled dataset.

**Definition 3.2** (Token–Semantic Alignment). For a vocabulary token $v$ with embedding $E_v$, we define its alignment with the expected semantic direction at position $(b, i)$ as

$$\text{Align}(b, i, v) = \cos(\bar{h}_{b,i}^{(s)}, E_v) = \frac{\langle \bar{h}_{b,i}^{(s)}, E_v \rangle}{\|\bar{h}_{b,i}^{(s)}\| \|E_v\|}. \quad (8)$$

**Proposition 3.3** (Prototype-Induced Logit Correction). *Let $\kappa(s)$ be a scheduling factor and $\lambda > 0$ a small correction weight. The prototype-conditioned logit update is*

$$\Delta\ell_{b,i,v}^{(s)} = \kappa(s)\,\text{sim}_b^{(s)}\,\text{Align}(b, i, v), \quad \ell^{(s)} \leftarrow \ell^{(s)} + \lambda\,\Delta\ell^{(s)}. \quad (9)$$

The term $\text{sim}_b$ evaluates whether the current image trajectory agrees with the prototype $p_{c,s}$; positive similarity indicates movement toward the prototype manifold. The term $\text{Align}(b, \ell, v)$ measures whether token $v$ continues this semantic direction. Their product is positive exactly when token $v$ reinforces prototype-aligned semantics. The small update scale $\lambda \ll 1$ ensures that VAR stability is preserved.

**Direction-Aware Guidance vs. Uniform Biasing.** Uniformly shifting logits by prototype similarity,

$$\Delta\ell_{b,\ell,v}^{(s)} = \kappa(s)\,\text{sim}_b, \quad (10)$$

fails to alter the probability distribution after softmax normalization and may increase the likelihood of prototype-inconsistent tokens. In contrast, the direction-aware update in Theorem 3.3 satisfies

$$\Delta\ell_{b,\ell,v}^{(s)} \propto \begin{cases} +1, & \text{if } v \text{ moves toward the prototype manifold,} \\ -1, & \text{if it moves away from the manifold,} \end{cases}$$

thus selectively strengthening prototype-consistent directions, suppressing inconsistent ones, and preserving diversity for neutral tokens.

**Stage-Aware Guidance.** We adopt a Gaussian schedule

$$\kappa(s) = \exp\left(-\frac{(\rho_s - 0.5)^2}{2\sigma^2}\right), \quad \rho_s = s/S, \quad (11)$$

which applies the strongest guidance at the semantic consolidation phase (mid-resolution) while preserving the stability of coarse layout initialization and late fine-detail refinement.

The refinement mechanism in Theorem 3.3 requires only prototype features and expected embeddings; it avoids enumerating the entire vocabulary and is therefore compatible with large-token VAR models. Our refinement strategy: 1) provides a differentiable semantic estimation without decoding all tokens, 2) aligns VAR sampling with prototype manifolds in a direction-aware manner, 3) imposes minimal computational overhead, and 4) preserves sampling diversity while enhancing semantic fidelity. We present a thorough theoretical justification of the proposed Prototype-Guided Logit Refinement in Section B.

### 3.3. Target-Guided Filtering

Under VAR's efficient generation process, we first draw a large candidate pool $P$ of synthetic images. Since generation may drift away from the target class, we introduce a lightweight teacher-guided filtering stage before prototype-based coreset construction.

**Definition 3.4.** For class $c$, the subset of $P$ on label $c$ is $P_c := \{(x_i, y_i) \in P \mid y_i = c\}$. Each image $x_i \in P_c$ re-

**Algorithm 1** Prototype-Guided VAR

---

**Input:** Training set $\mathcal{D}$, class set $\mathcal{C}$
**Required:** VAR $G$, feature extractor $\Phi$, teacher $\phi_{\theta_T}$
**Initialize:** $\mathcal{D}_{\text{syn}} \leftarrow \emptyset$

1: **for** each class $c \in \mathcal{C}$ **do**
2:    **Prototype construction:**
3:    $p_{c,1} = \arg\min_{x_i \in \mathcal{D}_c} \|\Phi(x_i) - \bar{\Phi}_c\|$
4:    $p_{c,k} = \arg\max_{x_i \in \mathcal{D}_c} \min_{j<k} \|\Phi(x_i) - \Phi(p_{c,j})\|$
5:    **Prototype-guided VAR sampling:**
6:    **for** stage $s \in \mathcal{S}_{\text{guide}}$ **do**
7:       Apply prototype-induced logit correction:
        $\ell^{(s)} \leftarrow \ell^{(s)} + \lambda \, \kappa(s) \, \text{sim}^{(s)} \cdot \text{Align}(\bar{h}^{(s)}, E)$
8:    **end for**
9:    **Target-guided filtering:**
10:    Compute teacher confidence:
      $\gamma_i = -\mathcal{L}(\phi_{\theta_T}(x_i), y_i), \ (x_i, y_i) \in P_c$
11:    Obtain confident pool $\hat{P}_c$ via Eq. 12
12:    Select IPC samples from $\hat{P}_c$ via Eq. 5
13:    $\mathcal{D}_{\text{syn}} \leftarrow \mathcal{D}_{\text{syn}} \cup \mathcal{I}_c$
14: **end for**
15: **return** $\mathcal{D}_{\text{syn}}$

---

ceives a confidence score $s_i$ from a teacher classifier. Based on these scores, we form a high-confidence subset $\hat{P}_c$.

We compute per-sample confidence using a fixed teacher network $\phi_{\theta_T}$. For each $(x_i, y_i) \in P_c$, the score is defined as $\gamma_i = -\mathcal{L}(\phi_{\theta_T}(x_i), y_i)$, where $\mathcal{L}$ is the cross-entropy loss. Larger scores indicate higher teacher confidence. We define the high-confidence subset by selecting all samples in $P_c$ whose scores belong to the top-$K$ values within the class:

$$\hat{P}_c = \left\{ (x_i, y_i) \in P_c \ \middle| \ \gamma_i \geq \gamma_{(K)} \right\}, \tag{12}$$

where $\gamma_{(K)}$ denotes the $K$-th largest score in the set $P_c$. If fewer than $K$ samples are correctly predicted by $\phi_{\theta_T}$, we fill remaining positions using the highest-scoring samples. Finally, for each class $c$, we apply the multi-scale prototype construction and coreset selection described in Section 3.1 on $\hat{P}_c$ to produce the final IPC images.

# 4. Experiments

**Datasets and Evaluation Metric.** To evaluate the effectiveness and generalization of our approach, we conduct experiments on the full ImageNet (Deng et al., 2009) and its commonly used subsets, including ImageWoof and ImageNette. For fair comparison, we follow the same experimental setting as in Wang et al. (2025). The distilled datasets are then evaluated across multiple architectures, including ResNet-18 (He et al., 2016), ResNet-50, ResNet-101, EfficientNet-B0 (Tan & Le, 2019), and MobileNet-V2 (Howard et al., 2017). Additionally, results on ImageNet-

A–E and ImageNet-100 are presented in Sections G and H.

**Baselines.** We compare our method with generative dataset distillation approaches and other state-of-the-art methods, including SRe$^2$L (Yin et al., 2023), D$^3$HR (Zhao et al., 2025), D$^4$M (Su et al., 2024), Minimax (Gu et al., 2024a), CaO$_2$ (Wang et al., 2025) and MGD$^3$ (Santiago et al., 2025). In addition, we evaluate two generative baseline models to demonstrate the effectiveness of generative approaches for dataset distillation.

**Implementation Details.** We adopt the pre-trained VAR-d30 model (Tian et al., 2024) with a resolution of 256×256 as the backbone. For prototype guidance, we set the guidance strength $\lambda$ to 0.015 and the standard deviation $\sigma$ to 0.15, resulting in an effective guidance interval of $[0.2, 0.8]$. This design is motivated by the observation that semantic structures in generative models are largely formed at intermediate steps (Yu et al., 2023), as further evidenced in Section C. In addition, we provide detailed analyses of the parameters in Sections I, L and M. All experiments are conducted on NVIDIA RTX 4090 or RTX A6000 GPUs.

## 4.1. Comparison with state-of-the-art methods

We compare our method with state-of-the-art approaches on multiple datasets and architectures, with results shown in Table 1. Firstly, we present the validation results on the challenging ImageWoof subset and the relatively simpler ImageNette subset in the first two sections of the table. As shown, our method achieves the best performance under IPC = 10, 50, and 100 when evaluated on ResNet-18 and ResNet-101. Notably, under IPC = 10, our method outperforms the current best approach by 1.6% on ImageWoof and 3.0% on ImageNette when tested with ResNet-18. Our method also attains the best or second-best performance on ResNet-50. The consistent improvements across different datasets and evaluation architectures demonstrate the effectiveness and strong generalization ability of the proposed method.

We further scale up our evaluation to ImageNet-1K, with the results summarized in the lower part of Table 1. Under the IPC 10 setting, our method consistently surpasses the previous state of the art by 0.7%, 0.5%, and 0.5% when using ResNet-18, ResNet-50, and ResNet-101, respectively. More importantly, the proposed method demonstrates consistent gains across all settings, leading to an average accuracy improvement of 0.4%, which highlights its robustness and scalability on large-scale benchmarks.

**Computational Cost.** As discussed in Section 1, VAR generation provides faster inference than DiT, with approximately 70× speedup. As shown in Table 1, despite its substantially higher generation speed, VAR still maintains strong generation quality compared to DiT. Benefiting from this, our method achieves a trade-off between performance

*Table 1.* Performance comparison with pre-trained diffusion and VAR models and other state-of-the-art methods on ImageNet and its subsets. The best, second-best, and third-best results are highlighted in  dark orange ,  orange , and  light orange , respectively.

| D | IPC | Test Model | DiT | VAR | SRe$^2$L | D$^3$HR | D$^4$M | Minimax | CaO$_2$ | MGD$^3$ | Ours |
|---|---|---|---|---|---|---|---|---|---|---|---|
| ImageWoof | 10 (0.8%) | ResNet-18 | 38.6$_{\pm0.4}$ | 38.8$_{\pm0.4}$ | 20.2$_{\pm0.2}$ | 41.1$_{\pm0.8}$ | 37.5$_{\pm1.8}$ | 40.1$_{\pm1.0}$ | 45.6$_{\pm1.4}$ | 42.9$_{\pm1.5}$ | 47.2$_{\pm1.3}$ |
| | | ResNet-50 | 35.9$_{\pm0.6}$ | 36.1$_{\pm0.9}$ | 17.3$_{\pm1.7}$ | 38.5$_{\pm1.2}$ | 35.0$_{\pm1.4}$ | 37.3$_{\pm1.1}$ | 40.1$_{\pm0.1}$ | 38.7$_{\pm1.2}$ | 42.3$_{\pm0.9}$ |
| | | ResNet-101 | 31.7$_{\pm0.6}$ | 31.4$_{\pm0.7}$ | 17.7$_{\pm0.9}$ | 35.1$_{\pm1.3}$ | 31.2$_{\pm1.9}$ | 34.2$_{\pm1.7}$ | 36.5$_{\pm1.4}$ | 36.3$_{\pm1.3}$ | 40.1$_{\pm1.0}$ |
| | 50 (3.8%) | ResNet-18 | 63.6$_{\pm1.1}$ | 62.8$_{\pm1.0}$ | 23.3$_{\pm0.3}$ | 70.3$_{\pm0.9}$ | 65.7$_{\pm1.7}$ | 67.0$_{\pm1.8}$ | 68.9$_{\pm1.1}$ | 67.6$_{\pm1.4}$ | 72.0$_{\pm0.7}$ |
| | | ResNet-50 | 60.1$_{\pm1.3}$ | 59.2$_{\pm0.8}$ | 24.8$_{\pm0.7}$ | 67.8$_{\pm1.3}$ | 62.2$_{\pm1.2}$ | 64.3$_{\pm0.9}$ | 68.2$_{\pm1.1}$ | 66.9$_{\pm1.2}$ | 68.0$_{\pm1.5}$ |
| | | ResNet-101 | 59.5$_{\pm1.3}$ | 58.1$_{\pm1.2}$ | 21.2$_{\pm0.2}$ | 63.0$_{\pm1.5}$ | 61.4$_{\pm1.5}$ | 62.7$_{\pm1.6}$ | 63.1$_{\pm1.3}$ | 62.6$_{\pm1.2}$ | 64.6$_{\pm1.1}$ |
| | 100 (7.7%) | ResNet-18 | 65.3$_{\pm0.5}$ | 64.2$_{\pm1.7}$ | 31.2$_{\pm2.2}$ | 72.5$_{\pm0.6}$ | 71.5$_{\pm1.3}$ | 71.2$_{\pm1.1}$ | 73.0$_{\pm0.7}$ | 74.1$_{\pm0.7}$ | 75.5$_{\pm0.6}$ |
| | | ResNet-50 | 64.2$_{\pm1.6}$ | 62.5$_{\pm1.1}$ | 29.6$_{\pm1.8}$ | 70.9$_{\pm1.7}$ | 70.7$_{\pm1.6}$ | 70.3$_{\pm1.5}$ | 71.6$_{\pm0.7}$ | 71.5$_{\pm0.9}$ | 73.1$_{\pm0.3}$ |
| | | ResNet-101 | 62.7$_{\pm1.4}$ | 61.0$_{\pm1.2}$ | 27.5$_{\pm1.9}$ | 70.1$_{\pm0.9}$ | 69.3$_{\pm1.5}$ | 69.2$_{\pm1.3}$ | 70.3$_{\pm0.6}$ | 70.8$_{\pm0.6}$ | 72.6$_{\pm0.4}$ |
| ImageNette | 10 (0.8%) | ResNet-18 | 58.2$_{\pm0.7}$ | 58.3$_{\pm0.5}$ | 29.4$_{\pm3.0}$ | 62.8$_{\pm0.7}$ | 59.3$_{\pm1.4}$ | 61.4$_{\pm0.7}$ | 65.0$_{\pm0.7}$ | 66.1$_{\pm2.1}$ | 68.0$_{\pm0.9}$ |
| | | ResNet-50 | 63.4$_{\pm0.6}$ | 64.1$_{\pm0.7}$ | 49.8$_{\pm2.1}$ | 66.8$_{\pm0.6}$ | 63.5$_{\pm1.3}$ | 66.4$_{\pm0.4}$ | 67.5$_{\pm0.8}$ | 67.8$_{\pm1.5}$ | 69.1$_{\pm1.4}$ |
| | | ResNet-101 | 52.9$_{\pm0.7}$ | 52.1$_{\pm0.4}$ | 23.4$_{\pm0.8}$ | 63.4$_{\pm1.4}$ | 60.6$_{\pm1.4}$ | 55.4$_{\pm4.5}$ | 66.3$_{\pm1.3}$ | 66.5$_{\pm1.6}$ | 68.7$_{\pm1.3}$ |
| | 50 (3.8%) | ResNet-18 | 79.3$_{\pm0.8}$ | 78.1$_{\pm1.5}$ | 40.9$_{\pm0.3}$ | 84.6$_{\pm1.2}$ | 82.5$_{\pm1.4}$ | 84.1$_{\pm0.2}$ | 84.5$_{\pm0.6}$ | 85.0$_{\pm1.5}$ | 85.8$_{\pm0.8}$ |
| | | ResNet-50 | 73.5$_{\pm1.1}$ | 72.5$_{\pm0.9}$ | 71.2$_{\pm0.3}$ | 82.1$_{\pm1.6}$ | 75.3$_{\pm0.9}$ | 77.1$_{\pm0.7}$ | 82.7$_{\pm0.3}$ | 82.2$_{\pm0.8}$ | 82.5$_{\pm0.5}$ |
| | | ResNet-101 | 71.9$_{\pm1.2}$ | 70.8$_{\pm1.3}$ | 36.5$_{\pm0.7}$ | 80.3$_{\pm1.7}$ | 74.6$_{\pm1.3}$ | 77.4$_{\pm0.8}$ | 81.7$_{\pm1.0}$ | 81.6$_{\pm1.2}$ | 82.3$_{\pm0.7}$ |
| | 100 (7.7%) | ResNet-18 | 81.4$_{\pm0.3}$ | 80.6$_{\pm0.7}$ | 52.3$_{\pm1.8}$ | 87.6$_{\pm0.5}$ | 87.1$_{\pm0.7}$ | 86.5$_{\pm1.2}$ | 88.2$_{\pm0.6}$ | 87.8$_{\pm0.9}$ | 89.1$_{\pm0.5}$ |
| | | ResNet-50 | 79.9$_{\pm0.4}$ | 78.5$_{\pm0.3}$ | 50.2$_{\pm1.6}$ | 86.9$_{\pm0.6}$ | 86.3$_{\pm1.2}$ | 84.6$_{\pm0.9}$ | 87.5$_{\pm0.8}$ | 87.1$_{\pm0.5}$ | 88.6$_{\pm0.5}$ |
| | | ResNet-101 | 77.2$_{\pm0.4}$ | 76.3$_{\pm0.5}$ | 46.7$_{\pm2.1}$ | 86.2$_{\pm0.8}$ | 84.7$_{\pm1.1}$ | 83.1$_{\pm1.7}$ | 86.6$_{\pm0.5}$ | 86.3$_{\pm0.7}$ | 88.0$_{\pm0.3}$ |
| ImageNet-1K | 10 (0.8%) | ResNet-18 | 39.8$_{\pm0.3}$ | 39.2$_{\pm0.5}$ | 21.3$_{\pm0.6}$ | 44.3$_{\pm0.3}$ | 27.9$_{\pm0.1}$ | 44.3$_{\pm0.5}$ | 46.1$_{\pm0.2}$ | 45.6$_{\pm0.5}$ | 46.8$_{\pm0.3}$ |
| | | ResNet-50 | 46.5$_{\pm0.5}$ | 45.1$_{\pm1.2}$ | 28.4$_{\pm0.1}$ | 51.5$_{\pm0.3}$ | 33.5$_{\pm0.1}$ | 49.7$_{\pm0.8}$ | 53.0$_{\pm0.2}$ | 52.3$_{\pm0.3}$ | 53.5$_{\pm0.5}$ |
| | | ResNet-101 | 42.7$_{\pm0.3}$ | 42.6$_{\pm0.6}$ | 30.9$_{\pm0.1}$ | 52.1$_{\pm0.4}$ | 34.2$_{\pm0.2}$ | 46.9$_{\pm1.3}$ | 52.2$_{\pm1.1}$ | 52.0$_{\pm0.8}$ | 52.7$_{\pm1.0}$ |
| | 50 (3.8%) | ResNet-18 | 52.9$_{\pm0.2}$ | 52.3$_{\pm0.5}$ | 46.8$_{\pm0.2}$ | 59.4$_{\pm0.1}$ | 55.2$_{\pm0.6}$ | 58.6$_{\pm0.3}$ | 60.0$_{\pm0.0}$ | 60.2$_{\pm0.1}$ | 60.3$_{\pm0.1}$ |
| | | ResNet-50 | 58.2$_{\pm0.1}$ | 58.0$_{\pm0.3}$ | 55.6$_{\pm0.3}$ | 64.0$_{\pm0.3}$ | 62.6$_{\pm0.6}$ | 64.8$_{\pm0.1}$ | 65.5$_{\pm0.1}$ | 64.6$_{\pm0.4}$ | 65.8$_{\pm0.3}$ |
| | | ResNet-101 | 59.1$_{\pm0.3}$ | 58.8$_{\pm0.6}$ | 60.8$_{\pm0.5}$ | 66.1$_{\pm0.1}$ | 63.4$_{\pm0.1}$ | 65.6$_{\pm0.1}$ | 66.2$_{\pm0.1}$ | 67.7$_{\pm0.4}$ | 67.9$_{\pm0.3}$ |
| | 100 (7.7%) | ResNet-18 | 54.7$_{\pm0.1}$ | 54.2$_{\pm0.1}$ | 52.8$_{\pm0.3}$ | 62.5$_{\pm0.0}$ | 59.3$_{\pm0.0}$ | 58.1$_{\pm0.1}$ | 62.1$_{\pm0.3}$ | 62.7$_{\pm0.1}$ | 63.0$_{\pm0.2}$ |
| | | ResNet-50 | 60.8$_{\pm0.0}$ | 61.0$_{\pm0.1}$ | 58.9$_{\pm0.4}$ | 65.7$_{\pm0.1}$ | 63.0$_{\pm0.2}$ | 62.8$_{\pm0.3}$ | 66.5$_{\pm0.2}$ | 66.3$_{\pm0.1}$ | 67.7$_{\pm0.1}$ |
| | | ResNet-101 | 63.2$_{\pm0.1}$ | 63.0$_{\pm0.0}$ | 62.8$_{\pm0.2}$ | 68.1$_{\pm0.0}$ | 66.5$_{\pm0.0}$ | 65.9$_{\pm0.3}$ | 68.5$_{\pm0.1}$ | 68.8$_{\pm0.1}$ | 69.6$_{\pm0.2}$ |

and efficiency. Notably, our approach does not rely on additional fine-tuning or surrogate objectives computation in the latent space. Instead, we directly guide VAR generation using prototypes. As illustrated in Figure 1, even after excluding the time required for Minimax fine-tuning, our method remains faster than Minimax while delivering superior performance. These results demonstrate that our method maintains high efficiency without sacrificing performance.

### 4.2. Ablation Study

**Component Analysis.** To evaluate the contribution of each component, we conduct an ablation study on ImageWoof and ImageNette with IPC = 10 and 50, using ResNet18 and ResNet50 as evaluation backbones. The results are summarized in Table 2. Without any guidance, VAR produces task-agnostic samples, leading to suboptimal performance. Introducing Prototype-Guided generation consistently improves accuracy across all datasets and IPC regimes. This confirms that the proposed multi-prototype feature bank provides informative and diverse semantic anchors for VAR, enhancing the representativeness of generated samples while avoiding mode collapse. Applying Task-Guided filtering alone also yields notable gains by removing misaligned samples and selecting IPC-aware representatives, indicating the importance of task-level supervision in constructing a

*Table 2.* Ablation study on different module combinations.

| Prototype-G. | Target-G. | ImageWoof | | ImageNette | |
|---|---|---|---|---|---|
| | | IPC10 | IPC50 | IPC10 | IPC50 |
| - | - | $38.8_{\pm0.4}$ | $62.8_{\pm1.0}$ | $58.3_{\pm0.5}$ | $78.1_{\pm1.5}$ |
| ✓ | - | $43.2_{\pm0.7}$ | $69.5_{\pm1.1}$ | $65.3_{\pm0.8}$ | $82.1_{\pm1.4}$ |
| - | ✓ | $42.6_{\pm0.6}$ | $68.9_{\pm0.8}$ | $64.1_{\pm0.9}$ | $81.2_{\pm0.9}$ |
| ✓ | ✓ | $47.2_{\pm1.3}$ | $72.0_{\pm0.7}$ | $68.0_{\pm0.9}$ | $85.8_{\pm0.8}$ |

*Table 3.* Ablation study on the filtering mechanism. "w/o pool" denotes direct generation, while "rand" denotes random selection from a $5\times$ pool.

| Method | ImageWoof | | ImageNette | |
|---|---|---|---|---|
| | IPC 10 | IPC 50 | IPC 10 | IPC 50 |
| Minimax | $40.1_{\pm1.0}$ | $67.0_{\pm1.8}$ | $61.4_{\pm0.7}$ | $84.1_{\pm0.2}$ |
| $D^3HR$ | $41.1_{\pm0.8}$ | $70.3_{\pm0.9}$ | $62.8_{\pm0.7}$ | $84.6_{\pm1.2}$ |
| Ours (w/o pool) | $43.2_{\pm0.7}$ | $69.5_{\pm1.1}$ | $65.3_{\pm0.8}$ | $82.1_{\pm1.4}$ |
| Ours (rand) | $45.1_{\pm1.1}$ | $70.6_{\pm1.3}$ | $66.3_{\pm1.3}$ | $84.2_{\pm1.5}$ |

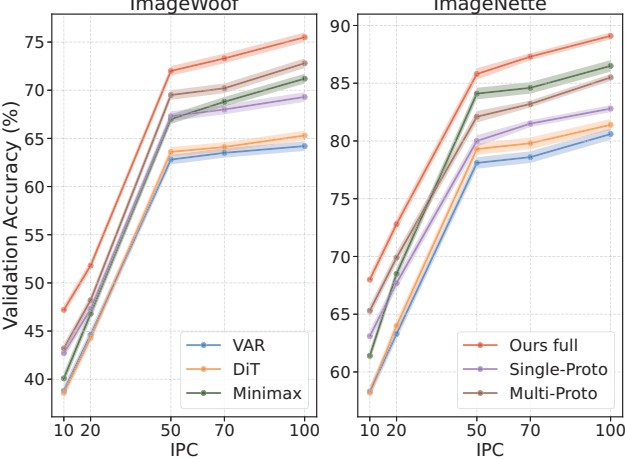

*Figure 3.* Guided by prototypes, the proposed method significantly enhances the diversity of generated images while maintaining their representativeness. As a result, it consistently outperforms baseline VAR models by a substantial margin across different IPC settings.

compact surrogate dataset. However, its improvement is relatively limited compared to prototype guidance, as it does not directly influence the generation process. Combining both components achieves the best performance. Prototype guidance improves the quality and diversity of generated samples at the source, while task-guided filtering further refines them to better align with downstream objectives. Their synergy results in a compact yet highly discriminative surrogate dataset, leading to substantial improvements under both low- and high-IPC settings.

**Effect of Prototype-Guided Generation.** We compare the proposed prototype-guided logit refinement with the baseline VAR, diffusion models (DiT), and DiT with Minimax fine-tuning to validate its effectiveness, as shown in Figure 3. Experiments are conducted on ImageWoof and ImageNette, representing challenging and relatively easy tasks, respectively. As discussed earlier, VAR is significantly faster than DiT. However, under small IPC settings (IPC = 10 and 20), VAR already generates informative samples, achieving validation performance comparable to DiT on both datasets. As IPC increases, a light performance gap emerges between VAR and diffusion-based methods, particularly when IPC exceeds 50. This indicates that unguided VAR generation lacks the capacity to maintain both representativeness and diversity as the surrogate dataset grows.

Introducing single-prototype-guided VAR significantly improves performance, especially under small IPCs. By steering generation toward a representative prototype, the model produces samples that are both informative and moderately diverse, which is particularly effective when only a few samples per class are allowed. Nevertheless, as IPC increases, the performance gain gradually saturates. As illustrated in Figure 4, the diversity of generated samples depends on the number of guiding prototypes. Under a single-prototype constraint, sample diversity remains limited, causing the generation process to overly rely on a dominant representative mode. In contrast, multi-prototype guidance further constrains the generation process toward multiple representative anchors, substantially enhancing sample diversity. Under small IPCs, the generated images contain richer and more discriminative information. More importantly, under larger IPCs, the generated dataset is encouraged to approach multiple prototypes, thereby preserving diversity while maintaining class representativeness. Compared to the unconstrained VAR generation shown in Figure 4, our method achieves a better balance between representativeness and diversity. Finally, combining prototype-guided generation with task-guided filtering consistently yields additional improvements, leading to the best overall performance. These results demonstrate that the proposed approach effectively mitigates both the limited diversity caused by prototype selection and the target-agnostic nature of generative methods, resulting in a compact yet reliable surrogate dataset.

**Effect of Target-Guided Filtering.** The target-guided filtering is not merely an auxiliary component, but rather a core capability uniquely unlocked by the inherent efficiency of VAR. Constructing an expansive candidate pool such as generating approximately 250,000 high-resolution images for a 5x pool on ImageNet-1K at IPC 50 is computationally prohibitive for diffusion models due to their iterative denoising process. However, VAR's parallel prediction accelerates generation by approximately 70x. This efficiency renders large-scale generative filtering highly viable for ProtoVAR, introducing a robust paradigm that remains fundamentally impractical for standard diffusion-based distillation.

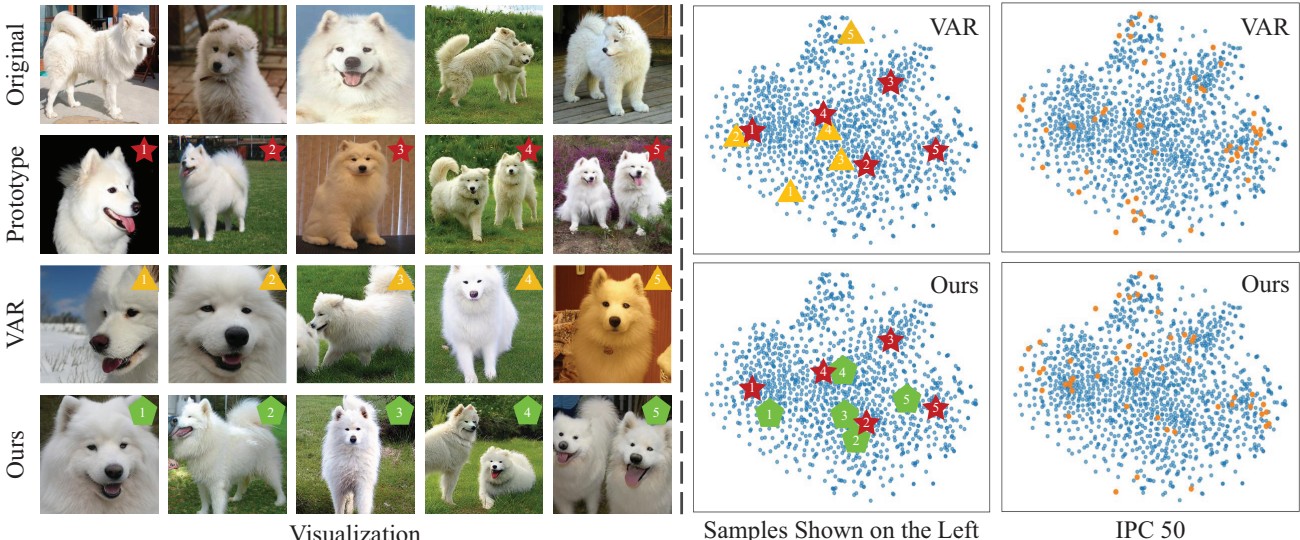

*Figure 4.* **Left**: Visualization of randomly selected original images, class prototypes, and synthetic images generated by the baseline model (VAR) and our proposed method. **Right**: t-SNE visualization of original samples, class prototypes, and synthetic samples produced by VAR and our method. From the first t-SNE column, which corresponds to the samples shown on the left, our method, guided by class prototypes, achieves higher diversity while maintaining representativeness compared to VAR. Consistent with the image visualization and the t-SNE results at IPC = 50, our method significantly improves the coverage of the original data distribution and enhances the diversity of the surrogate dataset.

To isolate the efficacy of our prototype-guided generation from the teacher classifier, we evaluate ProtoVAR under two unfiltered settings: (i) w/o pool (direct generation of the exact IPC budget) and (ii) rand (random selection from a 5x candidate pool). As shown in Table 3, ProtoVAR consistently outperforms other methods at IPC 10 and remains highly competitive at IPC 50, entirely independent of the filtering stage. The marginal performance shift at IPC 50 is inherently expected; while prototype guidance guarantees representativeness, sampling from a larger generated pool is necessary to maximize diversity bounds. Crucially, Proto-VAR establishes these robust baselines at a fraction of the computational cost required by diffusion models.

### 4.3. Visualization

**Generated Sample Comparison.** The proposed method substantially improves both the representativeness and diversity of the generated surrogate dataset. To explicitly illustrate these improvements, we compare samples generated using our method, the same backbone and prototype selection strategy in Figure 4. Images selected via prototypes exhibit high representativeness, as reflected in the t-SNE visualization on the right, where the prototypes are evenly distributed over the original dataset. However, since the prototypes are still drawn from the original dataset and are far fewer in number, their diversity is inherently limited, making it difficult to capture information beyond the original data. In contrast, samples generated by the baseline VAR model tend to share similar poses and predominantly capture the most salient features of objects. The t-SNE visualization

further indicates that these samples are randomly scattered across the original dataset distribution, with limited control over diversity and representativeness.

By comparison, the proposed ProtoVAR criterion significantly enhances the diversity of generated images while preserving prototype representativeness. On one hand, the generated images contain more prototype-related content, and the t-SNE visualization shows that these samples are uniformly distributed around the prototypes. Prototype guidance thus ensures better coverage of the overall original distribution, capturing more representative features. On the other hand, VAR produces samples that convey more diverse information, and following task-related filtering, the generated dataset attains increased diversity without compromising quality. Finally, the t-SNE distribution of samples generated at IPC 50 by our method demonstrates more comprehensive coverage of the original data distribution compared with baseline VAR models, while maintaining consistent sample density. As a result, the surrogate dataset more faithfully represents the original large-scale dataset, leading to improved validation performance.

**Representativeness and Diversity Score.** While the previous section provides a visual comparison of generated samples along with t-SNE embeddings, it does not offer quantitative evaluation metrics. Following the approach (Santiago et al., 2025), we numerically measure both diversity and representativeness. Specifically, we compare the scores of a subset of classes for VAR, Minimax, MGD[3], and our method, as shown in Figure 5, where higher values indicate

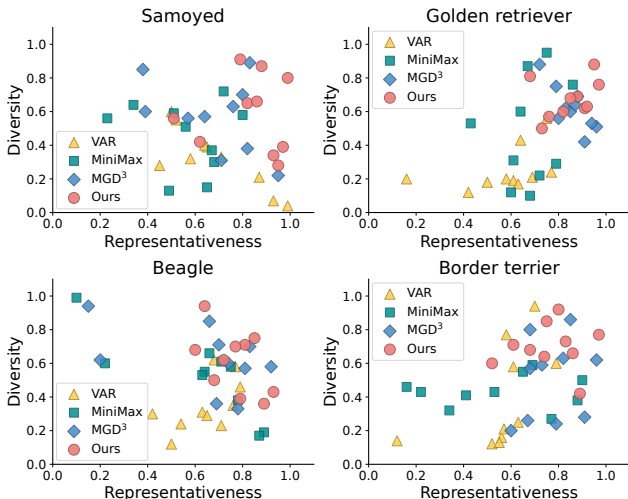

*Figure 5.* Comparison of representativeness and diversity for a subset of classes in the distilled dataset obtained using different methods on 10 IPC of ImageWoof. Each point represents an individual image in the distilled dataset.

greater representativeness and diversity. As observed, VAR achieves relatively high representativeness but exhibits limited diversity. Minimax diffusion generates more diverse samples than DiT, whereas MGD[3] demonstrates stronger representativeness, though its diversity can be constrained in certain cases. In contrast, our method attains a more balanced trade-off between diversity and representativeness, substantially enhancing the validation performance of the generated surrogate dataset.

## 5. Conclusion

In this work, we first analyze the strengths and limitations of coreset selection and existing diffusion-based dataset distillation methods, including limited diversity, indirect matching objectives, and target-agnostic generation. To address these issues, we propose a prototype-guided generation framework built upon VAR, which effectively enhances sample diversity while maintaining class-specific representativeness. Extensive experiments across multiple benchmarks and backbone architectures demonstrate that our method achieves state-of-the-art performance. Notably, our approach consistently outperforms prior methods without requiring additional fine-tuning and operates with improved efficiency. Furthermore, we conduct comprehensive component analyses and validate the effectiveness of the proposed method through both qualitative visualizations and quantitative ablation studies.

## Acknowledgements

The authors would like to thank the reviewers for their constructive feedback and valuable comments. This work was partially supported by the Zhejiang Province Natural Science Foundation of China under Grant LZ24F030004.

## Impact Statement

Efficient dataset distillation has the potential to make large-scale learning more practical and accessible by substantially reducing storage and computational requirements. By leveraging prototype-guided synthesis, ProtoVAR improves both accuracy and efficiency over diffusion-based approaches, enabling scalable, high-quality dataset distillation without relying on costly iterative denoising processes. This capability is particularly valuable in resource-constrained or privacy-sensitive scenarios, such as continual learning and federated learning. Overall, our work aims to advance the reliability and scalability of dataset distillation to support broader real-world deployment.

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

This appendix provides additional technical details, theoretical support, and extended experimental results to complement the main paper.

- Section A reviews related works that further contextualize our method.

- Section B presents the theoretical justification of the proposed Prototype-Guided Logit Refinement.

- Section C analyzes how different levels of prototype guidance influence model behavior.

- Section D analyzes the distinct requirements of DD compared to standard generative modeling.

- Section E describes the complete Prototype-Guided VAR algorithm.

- Section F evaluates the cross-architecture generalization ability of our approach.

- Sections G and H provide further results on ImageNet subsets and ImageNet-100, respectively.

- Section I studies the robustness of our approach under different coreset selection strategies.

- Section J provides further implementation details.

- Section K presents further ablation studies on ProtoVAR, focusing on target-guided filtering and model capacity.

- Sections L and M investigate the effect of key factors, including guidance strength and prototype configuration.

- Section N evaluates the robustness and generalization of our default hyperparameter settings across diverse datasets.

- Finally, Section O presents additional visualization results to give more intuitive insights.

## A. Related Works

### A.1. Dataset Distillation

Dataset distillation (DD) aims to compress the rich information contained in a large-scale dataset into a compact surrogate set, such that training on the distilled data can achieve performance comparable to training on the full dataset (Zhao et al., 2021; Zhao & Bilen, 2021). Beyond efficient training, distilled images have also proven beneficial in scenarios such as continual learning (Zhao & Bilen, 2023; Gu et al., 2024b) and privacy-preserving applications (Dong et al., 2022).

Most prior DD approaches follow a bi-level optimization paradigm and can be broadly categorized by the training metric they attempt to match. Typical objectives include matching training gradients (Zhao et al., 2021; Vahidian et al., 2025), feature distributions (Zhao & Bilen, 2023; Li et al., 2025), or training trajectories (Cazenavette et al., 2022; Cui et al., 2023). However, these bi-level methods incur substantial computational overhead, making them difficult to scale to large datasets such as ImageNet. To address this scalability issue, works such as SRe$^2$L (Yin et al., 2023) and G-VBSM (Shao et al., 2024) adopt a uni-level optimization framework: they synthesize images by aligning model outputs and BatchNorm statistics with a pretrained teacher model, and subsequently assign soft labels to the distilled data. RDED (Sun et al., 2024) improves efficiency via a diversity-driven strategy that directly operates on image crops and compositions. Despite their gains in efficiency, these methods still suffer from limited performance due to insufficiently expressive matching objectives.

### A.2. Generative Dataset Distillation

Recent works have also explored dataset distillation through generative models (Gu et al., 2024a; Zhao et al., 2025; Santiago et al., 2025). Minimax (Gu et al., 2024a) leverages pre-trained diffusion models and adapts them via representative and diversity-driven fine-tuning. D$^4$M (Su et al., 2024) builds upon Stable Diffusion by replacing random noise with structured noisy modes during sampling. CaO$_2$ (Wang et al., 2025) introduces a two-stage pipeline that combines sample selection with latent-space optimization.

Although effective, these approaches rely on precise latent-space selection or matching strategies, additional fine-tuning stages, or task-agnostic sampling procedures. In contrast, our method attains both high representativeness and strong task relevance without incurring such overhead.

# B. Theoretical Justification of Prototype-Guided Logit Refinement

**Notation Simplification.** In the following analysis, we omit the stage index $s$ for notational clarity. All derivations are conducted at a fixed VAR stage, and all quantities (e.g., logits $\ell$ and embeddings $\bar{h}$) are implicitly conditioned on the same stage unless otherwise specified. For notational simplicity, we drop the stage superscript and write $\ell \equiv \ell^{(s)}$. Accordingly, all quantities derived from the logits (e.g., $\Pi = \mathrm{softmax}(\ell)$ and $\bar{h}$) are implicitly defined at the same stage.

## B.1. Problem Setting and Key Challenge

Variational Autoregressive (VAR) image generation models synthesize images $x$ by sequentially predicting discrete multi-scale latent tokens in a coarse-to-fine manner. Specifically, a latent feature map $f \in \mathbb{R}^{h \times w \times C}$ is quantized into $S$ discrete token maps $R = (r_1, \ldots, r_S)$, where each $r_s \in [V]^{h_s \times w_s}$. The joint distribution factorizes as

$$P_\theta(R) = \prod_{s=1}^{S} P_\theta(r_s \mid r_{<s}), \tag{13}$$

and during the $s$-th step all $h_s w_s$ tokens are predicted in parallel conditioned on the previous scales $r_{<s}$. The final image $x$ is reconstructed from the multi-scale token maps $R$ through the decoder. This next-scale design preserves spatial locality, supports efficient block-wise causal attention and KV caching, and enables much faster inference than standard AR models, while retaining strong modeling capacity.

Our goal is to inject prototype-level semantic guidance into VAR sampling such that:

- the generated trajectory remains aligned with a target prototype manifold, and

- sampling diversity and autoregressive stability are preserved.

## B.2. Why Diffusion Is Ill-Suited for Prototype Guidance?

Diffusion models generate images by gradually denoising Gaussian noise. At intermediate timesteps, the latent states are mixtures of signal and noise, whose semantic content is inherently ambiguous and scale-dependent. Applying prototype guidance at these stages would require prototypes to represent class semantics consistently across different noise levels, which is fundamentally ill-defined.

As a result, prototype guidance in diffusion models faces a dilemma: early-stage guidance is unstable due to noise-dominated representations, while late-stage guidance has limited influence as the global structure has already been determined. As shown in Figure 6, directly applying prototype matching in diffusion models leads to severe visual distortions and category bias in the final samples. In particular, the generated images exhibit large-scale spatial deformations, collapsed textures, and partial semantic drift toward incorrect categories. This behavior suggests that due to the ambiguous semantics of noise-dominated intermediate states, early prototype guidance introduces unstable semantic bias that accumulates over timesteps, ultimately resulting in distorted or category-biased generations.

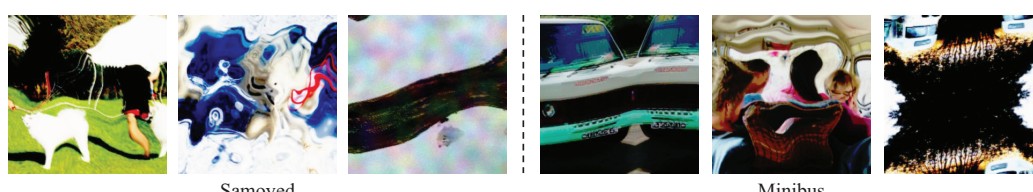

Samoyed                                                                 Minibus

*Figure 6.* Failure cases of diffusion models guided by direct prototype matching. Due to the ambiguous semantics of intermediate diffusion states, enforcing prototype-level alignment introduces unstable guidance, which accumulates over timesteps and leads to distorted or biased generations.

In contrast, VAR follows a coarse-to-fine autoregressive paradigm, where intermediate representations correspond to progressively refined image structures rather than noise-corrupted signals. This ensures that semantic features remain interpretable throughout decoding, making prototype alignment both well-posed and effective.

## B.3. Why Prototype Matching in Image or Feature Space is Inappropriate for AR Models?

A natural but flawed approach is to directly enforce prototype alignment by minimizing a feature-level distance, e.g.,

$$\min_x \|f(x) - p\|, \tag{14}$$

or to bias generation using gradients derived from this objective. However, this strategy is theoretically incompatible with AR generation for the following reasons.

**Non-differentiability of Token Sampling.** In AR models, the image is obtained via a sequence of discrete sampling steps:

$$v_t \sim \text{Categorical}(\text{softmax}(\ell_t)). \tag{15}$$

The mapping

$$\ell_t \;\mapsto\; v_t \;\mapsto\; x \tag{16}$$

is non-differentiable. Consequently, gradients of a prototype loss

$$\nabla_{\ell_t} \|f(x) - p\| \tag{17}$$

are either undefined or require surrogate relaxations that introduce severe bias.

## B.4. Expected Embedding as a Differentiable Semantic Proxy

To overcome these limitations, we introduce an expected embedding relaxation. Given logits $\ell_{b,i}$ at a token position, we define

$$\bar{h}_{b,i} = \mathbb{E}_{v \sim \Pi_{b,i,:}}[E_v] = \Pi_{b,i,:} E, \tag{18}$$

where $\Pi = \text{softmax}(\ell)$ denotes the token probability tensor. This serves as a continuous surrogate of the model's current semantic direction.

This construction has two crucial properties:

- **Differentiability.** $\bar{h}_{b,i}$ is a smooth function of the logits $\ell$, enabling principled logit-level refinement.

- **Local Semantic Fidelity.** $\bar{h}_{b,i}$ approximates the expected contribution of future token choices without committing to any single discrete token.

**Analysis of Expected Embedding Relaxation.** A potential concern is that the expected embedding $\bar{h}_{b,i}$ may not faithfully approximate the actual discrete autoregressive trajectory, since it averages token embeddings under a probabilistic distribution rather than selecting a single discrete token. Furthermore, tentative decoding from such soft representations could potentially introduce additional bias when passed through a decoder originally designed for discrete codebook entries.

To analyze this approximation gap, let $J(h)$ denote an arbitrary downstream semantic objective defined on token embeddings. The true expected objective under discrete autoregressive sampling is

$$\mathbb{E}_{v \sim \Pi}[J(E_v)], \tag{19}$$

while our method uses the expected embedding $\bar{h} = \mathbb{E}_{v \sim \Pi}[E_v]$ to compute a surrogate objective $J(\bar{h})$. Using a second-order Taylor expansion around $\bar{h}$ gives

$$\mathbb{E}_{v \sim \Pi}[J(E_v)] = J(\bar{h}) + \nabla J(\bar{h})^\top \mathbb{E}_{v \sim \Pi}[E_v - \bar{h}] + \frac{1}{2}\mathbb{E}_{v \sim \Pi}\left[(E_v - \bar{h})^\top \nabla^2 J(\xi_v)(E_v - \bar{h})\right], \tag{20}$$

where $\xi_v$ lies on the segment between $E_v$ and $\bar{h}$.

Crucially, the first-order term vanishes because

$$\mathbb{E}_{v \sim \Pi}[E_v - \bar{h}] = \bar{h} - \bar{h} = 0. \tag{21}$$

*Table 4.* Evolution of Top-1 token probability during autoregressive generation.

| Generation Step | 1 | 2 | 3 | 4 | 5 | 6 | 7 | 8 | 9 | 10 |
|---|---|---|---|---|---|---|---|---|---|---|
| **Top-1 Probability** | 0.92 | 0.62 | 0.79 | 0.82 | 0.85 | 0.87 | 0.90 | 0.95 | 0.97 | 0.99 |

Therefore, the discrepancy between the discrete objective and the continuous surrogate is governed purely by a second-order variance term.

Assuming that $J(\cdot)$ has an $L$-bounded Hessian, the approximation error satisfies

$$\left| \mathbb{E}_{v\sim\Pi}[J(E_v)] - J(\bar{h}) \right| \leq \frac{L}{2}\mathrm{Tr}(\Sigma), \tag{22}$$

where

$$\Sigma = \mathbb{E}_{v\sim\Pi}\left[ (E_v - \bar{h})(E_v - \bar{h})^\top \right] \tag{23}$$

is the covariance of the token embedding distribution.

This result provides a direct theoretical justification for the expected embedding relaxation. In a well-trained VAR model, autoregressive decoding progressively concentrates probability mass toward highly confident tokens. As the distribution sharpens, the covariance term $\mathrm{Tr}(\Sigma)$ rapidly decreases toward zero, causing the approximation error bound to vanish. Consequently, the expected embedding $\bar{h}$ becomes an increasingly accurate approximation of the underlying discrete generation trajectory.

Another concern is whether tentative decoding from soft embeddings introduces semantic distortion. Let $D(\cdot)$ denote the VQ-VAE decoder. Since $D$ is a continuous neural network, it satisfies a local Lipschitz condition:

$$\|D(\bar{h}) - D(E_v)\| \leq L_D\|\bar{h} - E_v\|, \tag{24}$$

meaning that deviations induced by soft embeddings remain bounded. In practice, this mainly introduces mild local smoothing rather than large-scale structural deformation.

Moreover, the tentatively decoded image is not directly used as the final generation output. Instead, it is immediately processed by a deep semantic feature extractor to compute prototype similarity. Such CNN-based feature extractors primarily capture global semantic structures and are inherently robust to small pixel-level perturbations. Therefore, minor soft-decoding artifacts are largely suppressed in the semantic feature space.

Architecturally, tentative decoding merely serves as an informational probe. If high uncertainty yields a blurry image, prototype similarity ($\mathrm{sim}_b$) drops to zero, naturally fading out the guidance. The actual intervention remains a minimal first-order perturbation ($\lambda \ll 1$) applied to the logits. The final generation trajectory never physically leaves the discrete network, guaranteeing perfectly sharp, artifact-free images.

We further validate this approximation empirically. First, we compute the cosine similarity between hard-decoded samples and soft-decoded samples in the ResNet50 feature space. The similarity consistently remains between $0.94$ and $0.98$ during the middle and late generation stages, indicating that the soft decoding trajectory preserves nearly identical macro-semantic structures as the discrete trajectory.

Second, we track the Top-1 probability throughout autoregressive decoding. As shown in Table 4, the token distribution rapidly collapses toward highly confident predictions as generation progresses, empirically confirming the variance shrinking behavior predicted by the above analysis.

### B.5. Logit-Level Guidance as First-Order Semantic Control

We now justify why prototype-guided logit refinement is both sufficient and optimal for AR generation.

**Proposition 1 (Logit Perturbation as First-Order Distribution Control).** Let $\pi(v) = \mathrm{softmax}(\ell)_v$. For a small logit perturbation $\Delta\ell$, the induced change in the token distribution satisfies

$$\Delta\pi(v) \approx \pi(v)\Big(\Delta\ell_v - \mathbb{E}_{v'}[\Delta\ell_{v'}]\Big). \tag{25}$$

**Implication.** Only relative, token-dependent logit corrections affect the sampling distribution. Uniform biasing is canceled by softmax normalization.

**Corollary (Failure of Uniform Prototype Biasing).** A uniform shift

$$\Delta \ell_{b,i,v} = \kappa(s) \operatorname{sim}_b \tag{26}$$

induces $\Delta \pi(v) = 0$ for all $v$, and therefore cannot selectively promote prototype-consistent tokens. This formally explains why naive prototype similarity biasing is ineffective.

As shown in the middle panel of Figure 7, uniform biasing shifts all logits equally, ignoring token- or feature-specific directions. As a result, the sampling distribution remains unchanged because the softmax normalization cancels out uniform shifts. This provides a geometric justification for Proposition 1 and its corollary: uniform prototype bias cannot selectively promote prototype-consistent tokens.

### B.6. Direction-Aware Logit Refinement

Our method introduces token-specific corrections:

$$\Delta \ell_{b,\ell,v} = \kappa(s) \operatorname{sim}_b \operatorname{Align}(b, i, v), \tag{27}$$

where

$$\operatorname{Align}(b, i, v) = \cos\!\big(\bar{h}_{b,i}, E_v\big). \tag{28}$$

**Proposition 2 (Semantic Monotonicity of Direction-Aware Updates).** Assume $\lambda \ll 1$. Then the expected semantic embedding after refinement satisfies

$$\mathbb{E}[\bar{h}_{b,i}^{\text{new}}] - \bar{h}_{b,i} \propto \operatorname{sim}_b \sum_v p(v) \operatorname{Align}(b, i, v)\, E_v. \tag{29}$$

**Interpretation.** When $\operatorname{sim}_b > 0$, probability mass is shifted toward tokens whose embeddings align with the current semantic trajectory, thereby moving the generation toward the prototype manifold in expectation.

### B.7. Geometric Interpretation of Prototype-Guided Logit Bias

We aim to impose a lightweight, stable, and semantically consistent guidance on the generation process using prototype information. Rather than treating the logit bias as a heuristic, we reinterpret it from a geometric constraint perspective and explain why such a design is principled.

**Prototype similarity from a geometric perspective.** Given an intermediate generated sample at block $b$, we extract its feature representation

$$\mathbf{f}(\hat{x}_b) \in \mathbb{R}^d, \tag{30}$$

and compare it with the class–scale prototype $\mathbf{p}_{c,s}$. We define their cosine similarity as

$$\operatorname{sim}_b = \cos\!\big(\mathbf{f}(\hat{x}_b), \mathbf{p}_{c,s}\big) = \frac{\langle \mathbf{f}(\hat{x}_b), \mathbf{p}_{c,s} \rangle}{\|\mathbf{f}(\hat{x}_b)\|\, \|\mathbf{p}_{c,s}\|}. \tag{31}$$

This quantity measures the angular alignment between the current generation state and the target prototype direction:

(i) if $\mathbf{f}(\hat{x}_b)$ moves toward the prototype manifold, the angle decreases and $\operatorname{sim}_b \to 1$;

(ii) if they are nearly orthogonal, $\operatorname{sim}_b \approx 0$;

(iii) if the directions are opposite, $\operatorname{sim}_b < 0$.

$\operatorname{sim}_b$ provides a differentiable signal indicating whether the generation trajectory already aligns with the desired semantic direction. Therefore, prototype guidance should be strengthened only when meaningful alignment has already emerged, rather than being enforced unconditionally. As shown on the left side of Figure 7, blindly forcing samples to move toward the prototype may lead to mode collapse, where semantic diversity is suppressed and samples collapse around a single mode, or even result in visually incorrect generations.

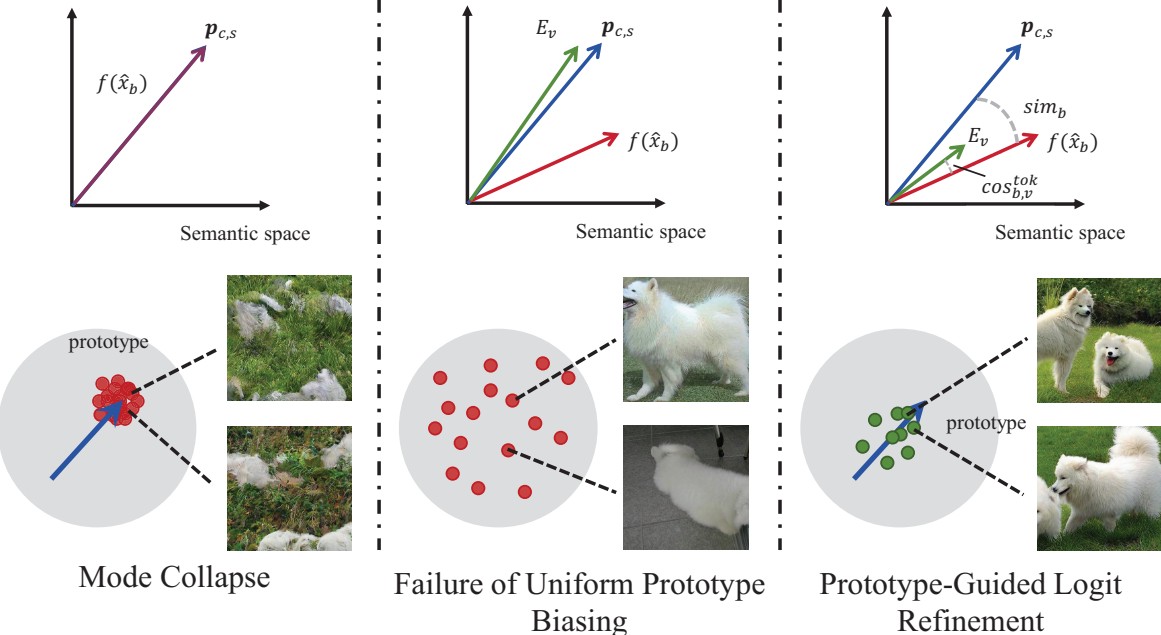

*Figure 7.* Effects of different prototype biasing strategies. **Left**: Unconditional attraction causes mode collapse and may produce erroneous samples. **Middle**: Uniform biasing equally shifts all logits, leaving the sampling distribution unchanged. **Right**: Our direction-aware refinement applies a small, direction-aware perturbation that guides samples toward the prototype while preserving diversity.

**Geometric meaning of token similarity.** In an autoregressive transformer, the logit for token $v$ at block $b$ is computed as

$$\ell_{b,v} = \langle \bar{\mathbf{h}}_b, \mathbf{E}_v \rangle, \tag{32}$$

where $\bar{\mathbf{h}}_b$ denotes the output embedding and $\mathbf{E}_v$ is the token embedding. We define the token-level cosine similarity as

$$\cos_{b,v}^{\text{tok}} = \frac{\bar{\mathbf{h}}_b^\top \mathbf{E}_v}{\|\bar{\mathbf{h}}_b\| \, \|\mathbf{E}_v\|}. \tag{33}$$

This term measures how well token $v$ aligns with the current semantic direction of the generation state.

Rather than uniformly boosting all tokens, guidance should be direction-aware, promoting tokens that are already compatible with the evolving semantics while suppressing incompatible ones.

**Joint logit bias as a dual directional constraint.** We combine the two directional signals into a unified logit bias:

$$\Delta\ell_{b,s,v} = \kappa(s) \, \text{sim}_b \, \cos_{b,v}^{\text{tok}}. \tag{34}$$

Equivalently,

$$\Delta\ell \propto \underbrace{\cos\big(\mathbf{f}(\hat{x}_b), \mathbf{p}_{c,s}\big)}_{\text{prototype alignment}} \times \underbrace{\cos\big(\bar{\mathbf{h}}_b, \mathbf{E}_v\big)}_{\text{semantic consistency}}. \tag{35}$$

$\text{sim}_b$ constrains the class–prototype direction, while $\cos_{b,v}^{\text{tok}}$ constrains the current semantic output direction. Their product ensures that strong bias is applied only when both constraints are satisfied, preventing over-forcing and preserving structural integrity during generation. As shown in the right panel of Figure 7, our direction-aware logit refinement introduces only a small perturbation in logit space, biasing the sampling distribution without enforcing hard alignment. Tokens are adjusted based on their consistency with both the current semantics and the class prototype, gently guiding samples toward the prototype while preserving diversity. This balances semantic control and variability, effectively preventing collapse.

**Gaussian scheduling from an information-dynamics viewpoint.** We adopt a Gaussian schedule to modulate the guidance strength:

$$\kappa(s) = \exp\left(-\frac{(\rho_s - 0.5)^2}{2\sigma^2}\right), \quad \rho_s = \frac{s}{S}. \tag{36}$$

The schedule peaks at intermediate stages ($\rho_s = 0.5$) and smoothly decays at early and late stages.

Early stages contain unstable, low-level structure where guidance is unreliable; middle stages capture the richest semantic information where guidance is most effective; late stages refine fine details where semantic forcing may be harmful. The Gaussian window thus approximates a temporal region of maximal semantic controllability.

**Sufficiency of small updates.** The final logit update is

$$\ell \leftarrow \ell + \lambda\Delta\ell, \quad \lambda \approx 0.015. \tag{37}$$

A small $\lambda$ ensures that the guidance acts as a first-order perturbation in logit space, preserving the topology of the original distribution.

We gently bias the generation direction rather than reshaping the entire distribution, leading to stable and natural prototype-aligned generation without mode collapse.

## C. Analysis of Prototype Guidance Effect

To evaluate how the strength of prototype guidance influences generation, we vary the guidance scale by adjusting $\sigma$ from 0.05 to 0.25, corresponding to an effective range expanding from $[0.4, 0.6]$ to nearly $[0, 1]$. A larger $\sigma$ indicates that guidance persists over a larger portion of the generation process. The upper part of Figure 8 reports how the evaluation metrics evolve under different $\sigma$ values, while quantitative analyses of representativeness and diversity are provided for $\sigma = 0.05$, 0.15, and 0.25. When $\sigma = 0.05$, the model preserves high diversity but exhibits relatively low representativeness, which aligns with the visual results in the lower part of Figure 8, where more diverse styles appear. In contrast, when guidance dominates almost the entire process (e.g., $\sigma = 0.25$), the outputs remain highly aligned with the prototype but lose diversity, producing structurally similar samples, as also evidenced by the visual results. From the metric trends, $\sigma = 0.15$ achieves the best overall performance, striking a good balance between representativeness and diversity, which is validated by both quantitative indicators and visual comparisons. Therefore, we adopt $\sigma = 0.15$ as the default setting.

## D. Analysis of Visual Expressiveness

In this section, we analyze the distinct requirements of DD compared to standard generative modeling. While typical generative models optimize for visual realism, DD requires visual expressiveness to provide effective training signals.

Diffusion-based approaches prioritize the data distribution's mean to achieve low FID. However, under constrained budgets (e.g., 10 or 50 IPC), this tendency often yields redundant, "average" samples that lack the diversity required to prevent student model overfitting. In contrast, the hierarchical nature of VAR allows for the injection of dense, discriminative class features at multiple scales without stochastic interference. This ensures that synthetic images, while remaining visually competitive, provide stronger and more accurate gradients for training. As evidenced in Table 5, ProtoVAR achieves superior visual fidelity compared to diffusion-based baselines while maintaining higher semantic discriminability. This demonstrates that scale-wise autoregressive generation is inherently more suitable for distilling information-dense datasets.

*Table 5.* Comparison of FID on ImageNet-1K. Lower FID indicates synthetic samples are closer to the original data distribution.

| Method | FID (10 IPC) | FID (50 IPC) |
|---|---|---|
| Minimax (Diffusion) | 18.6 | 15.0 |
| VAR (Standard) | 17.2 | 12.9 |
| ProtoVAR (Ours) | **17.0** | **13.1** |

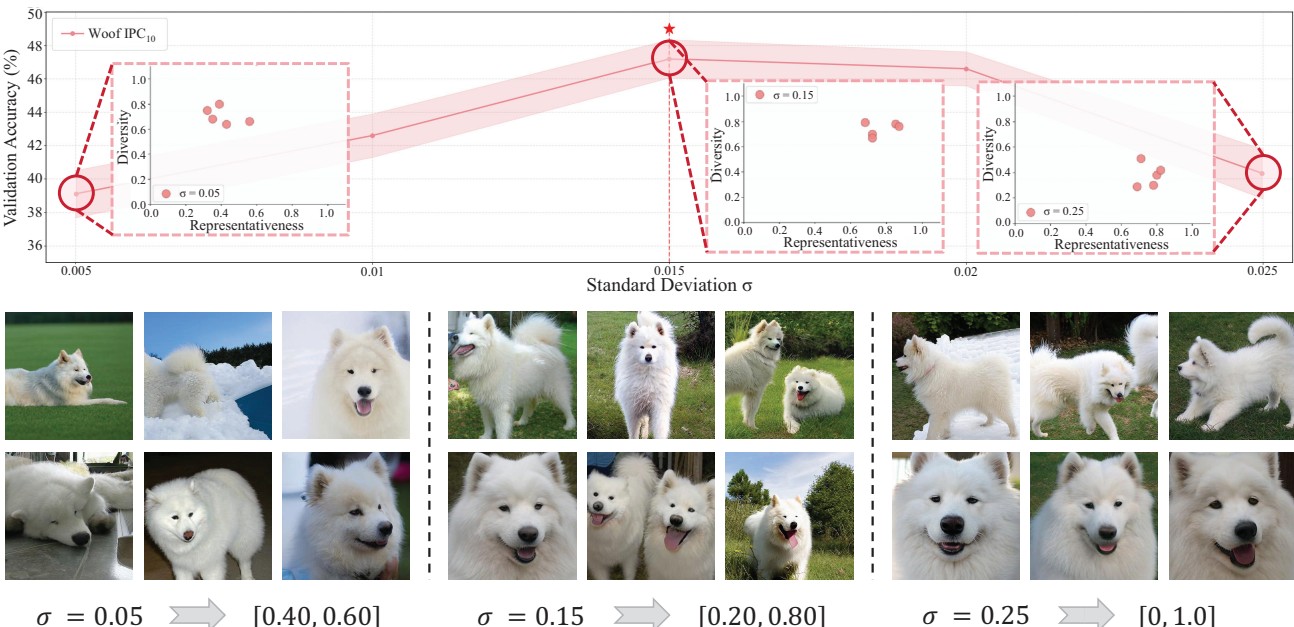

$\sigma = 0.05 \implies [0.40, 0.60]$   $\sigma = 0.15 \implies [0.20, 0.80]$   $\sigma = 0.25 \implies [0, 1.0]$

*Figure 8.* Effect of prototype guidance strength. **Top:** Evaluation metrics under different $\sigma$ values, showing how the guidance range affects the results, with corresponding analyses on representativeness and diversity. **Bottom:** Visual comparison at $\sigma = 0.05, 0.15,$ and 0.25. Smaller $\sigma$ yields higher diversity but weaker representativeness, while excessively large $\sigma$ leads to over-alignment and reduced diversity. $\sigma = 0.15$ achieves the best balance.

## E. Prototype-Guided VAR Algorithm

To clearly illustrate the proposed framework, Algorithm 2 summarizes the complete procedure of Prototype-Guided VAR. Given the training set $\mathcal{D}$ and class set $\mathcal{C}$, we first construct a compact yet representative prototype bank for each class. Specifically, we compute the mean feature $\bar{\Phi}_c$ using the feature extractor $\Phi$ and select an initial prototype closest to this mean. Additional prototypes are then iteratively chosen to maximize pairwise diversity, forming a multi-scale prototype bank that balances representativeness and coverage across different semantic granularities.

With the constructed prototypes, we perform prototype-guided sampling within the VAR generator $G$. At each guidance stage $s$, we compute prototype similarity and token-level semantic alignment, which jointly provide a structured and informative signal for steering generation. This signal is injected through a logit correction mechanism modulated by a stage-dependent weight $\kappa(s)$, allowing strong semantic guidance in mid-resolution stages while maintaining stable coarse layout initialization and fine detail refinement.

Finally, we introduce a target-guided filtering stage to enhance sample reliability. A teacher network $\phi_{\theta_T}$ evaluates generated candidates, and only confident samples are retained to form a high-quality pool. From this pool, IPC samples are selected via the coreset criterion again to construct the final distilled dataset $\mathcal{D}_{\text{syn}}$. This procedure ensures that the resulting synthetic data are not only representative and diverse, but also aligned with task-relevant semantics.

## F. Cross-architecture Generalization

To further evaluate cross-architecture performance, we compare our method with recent baselines under IPC 10 and IPC 50 on ImageNet-1K. As shown in Figure 9, our approach consistently achieves significant accuracy improvements across all evaluated backbone architectures. Notably, under prototype-guided generation, the samples synthesized by VAR are sufficient to deliver competitive and stable performance across diverse model architectures.

To further evaluate cross-model generalization, we conduct experiments on ImageNet-1K by incorporating a broader set of teacher architectures, including ResNet-50, ResNet-101, EfficientNet-B0, and ShuffleNet-V2. Here, the teacher model provides soft labels for validation. As shown in Table 6, our method consistently achieves superior performance across these substantially different architectures.

**Algorithm 2** Detailed Prototype-Guided VAR

---

**Input:** Training set $\mathcal{D}$, class set $\mathcal{C}$
**Required:** VAR $G$, feature extractor $\Phi$, teacher $\phi_{\theta_T}$
**Initialize:** $\mathcal{D}_{\text{syn}} \leftarrow \emptyset$

1: **for** each class $c \in \mathcal{C}$ **do**
2:     **Prototype construction:**
3:     Compute $\bar{\Phi}_c = \frac{1}{|\mathcal{D}_c|} \sum_{x_i \in \mathcal{D}_c} \Phi(x_i)$
4:     $p_{c,1} = \arg\min_{x_i \in \mathcal{D}_c} \|\Phi(x_i) - \bar{\Phi}_c\|$
5:     **for** $k = 2$ to $M$ **do**
6:         $p_{c,k} = \arg\max_{x_i \in \mathcal{D}_c} \min_{j<k} \|\Phi(x_i) - \Phi(p_{c,j})\|$
7:     **end for**
8:     Construct multi-scale bank $\{p_{c,s}\}_{s=1}^{S}$
9:     **Prototype-guided VAR sampling:**
10:     **for** stage $s \in \mathcal{S}_{\text{guide}}$ **do**
11:         Obtain prototype similarity sim via Eq. 7
12:         Obtain token semantic alignment Align via Eq. 8
13:         Apply prototype-induced logit correction:

$$\ell^{(s)} \leftarrow \ell^{(s)} + \lambda\,\kappa(s)\,\text{sim}^{(s)} \cdot \text{Align}(\bar{h}^{(s)}, E)$$

14:     **end for**
15:     **Target-guided filtering:**
16:     Compute teacher confidence:

$$\gamma_i = -\mathcal{L}(\phi_{\theta_T}(x_i), y_i),\ (x_i, y_i) \in P_c$$

17:     Obtain confident pool $\hat{P}_c$ via Eq. 12
18:     Select IPC samples from $\hat{P}_c$ via Eq. 5
19:     $\mathcal{D}_{\text{syn}} \leftarrow \mathcal{D}_{\text{syn}} \cup \mathcal{I}_c$
20: **end for**
21: **return** $\mathcal{D}_{\text{syn}}$

---

### F.1. Generalization to Vision Transformers

Most dataset distillation methods are mainly evaluated on CNNs, especially ResNets, due to the different training characteristics of Vision Transformers (ViTs). Unlike CNNs, ViTs lack strong spatial inductive biases and usually require much larger training data. When trained on highly compact distilled datasets (e.g., IPC 10 or IPC 50), ViTs are prone to severe overfitting. Therefore, CNNs are commonly adopted as the primary evaluation architecture to better measure the intrinsic quality of distilled data. Although CNNs serve as our main evaluation protocol, we additionally study the transferability of distilled knowledge to ViT under IPC 10 and IPC 50 for ImageWoof, ImageNette, and ImageNet-1K. The results are summarized in Table 7.

ProtoVAR consistently achieves better performance than Minimax across all settings. These results indicate that our prototype-guided generation captures richer, more robust semantic features transferable across diverse architectures.

## G. Further Evaluation on ImageNet Subsets

To further assess the generalization capability of our approach, we compare it with GLaD, H-GLaD, and LM3D under the same cross-architecture evaluation protocol. Following their setup, we conduct experiments using AlexNet, VGG11, ResNet18, and ViT, and report the mean performance by repeating each evaluation five times per architecture. We adopt the five ImageNet subsets (A, B, C, D, and E) defined in GLaD for testing. As shown in Table 8, our method consistently surpasses existing approaches across all subsets, demonstrating strong scalability and superior performance across diverse ImageNet subsets.

Table 6. Comparison of Top-1 accuracy for cross-architecture generalization on ImageNet-1K, IPC = 10.

| Student \ Teacher | | ResNet-18 | EfficientNet-B0 | MobileNet-V2 |
|---|---|---|---|---|
| ResNet-18 | RDED | $42.3_{\pm0.6}$ | $31.0_{\pm0.1}$ | $40.4_{\pm0.1}$ |
| | Minimax | $44.3_{\pm0.5}$ | $29.3_{\pm0.4}$ | $42.2_{\pm0.6}$ |
| | Ours | $46.8_{\pm0.3}$ | $31.2_{\pm0.1}$ | $45.2_{\pm0.8}$ |
| ResNet-50 | RDED | $43.6_{\pm0.5}$ | $32.7_{\pm1.2}$ | $40.6_{\pm0.6}$ |
| | Minimax | $49.7_{\pm0.8}$ | $35.2_{\pm1.3}$ | $46.8_{\pm0.7}$ |
| | Ours | $53.5_{\pm0.5}$ | $38.1_{\pm0.8}$ | $49.2_{\pm0.4}$ |
| ResNet-101 | RDED | $48.3_{\pm1.0}$ | $36.5_{\pm1.1}$ | $44.7_{\pm0.9}$ |
| | Minimax | $46.9_{\pm1.3}$ | $35.9_{\pm1.5}$ | $43.9_{\pm1.0}$ |
| | Ours | $52.7_{\pm1.0}$ | $39.2_{\pm1.0}$ | $47.6_{\pm0.8}$ |
| EfficientNet-B0 | RDED | $42.8_{\pm0.5}$ | $33.3_{\pm0.3}$ | $43.6_{\pm0.0}$ |
| | Minimax | $47.5_{\pm0.5}$ | $36.3_{\pm0.7}$ | $51.3_{\pm0.3}$ |
| | Ours | $52.2_{\pm0.3}$ | $41.6_{\pm0.5}$ | $53.6_{\pm0.2}$ |
| MobileNet-V2 | RDED | $34.4_{\pm0.2}$ | $31.2_{\pm0.8}$ | $38.5_{\pm0.6}$ |
| | Minimax | $41.7_{\pm0.6}$ | $30.2_{\pm1.0}$ | $43.3_{\pm0.5}$ |
| | Ours | $45.6_{\pm0.5}$ | $32.5_{\pm0.6}$ | $47.1_{\pm0.3}$ |

Table 7. Performance comparison on ViT trained with distilled datasets.

| Dataset | IPC | Minimax | Ours |
|---|---|---|---|
| ImageWoof | 10 | $26.5_{\pm2.3}$ | $\mathbf{35.1}_{\pm1.9}$ |
| | 50 | $38.2_{\pm1.7}$ | $\mathbf{46.1}_{\pm1.2}$ |
| ImageNette | 10 | $35.8_{\pm2.2}$ | $\mathbf{42.7}_{\pm2.3}$ |
| | 50 | $51.4_{\pm1.9}$ | $\mathbf{56.5}_{\pm1.8}$ |
| ImageNet-1K | 10 | $27.3_{\pm1.8}$ | $\mathbf{33.6}_{\pm1.1}$ |
| | 50 | $46.9_{\pm1.3}$ | $\mathbf{50.2}_{\pm0.9}$ |

# H. Further Evaluation on ImageNet-100

In addition to the 10-class ImageNet subsets and ImageNet-1K, we further evaluate our method on ImageNet-100, and the results are reported in Table 9. To ensure fair comparison, we follow the experimental setup of Gu et al. (2024a): the original resolution is 224×224, and all images are resized to 256×256 during training. Under the IPC setting of 10, the conventional dataset distillation method IDC-1 still achieves relatively strong performance. However, as IPC increases, our method consistently delivers stable improvements over existing approaches across different settings. Moreover, it is worth noting that our algorithm does not sacrifice efficiency for performance gains. As shown in Figure 1, it maintains high computational efficiency while achieving superior accuracy.

# I. Generalization Across Different Coreset Selection Strategies

To investigate the influence of different coreset selection algorithms and further assess the generalization ability of our approach, we evaluate several commonly used strategies, including random sampling, DBSCAN, $k$-means, and the prototype-based selection method. Experiments are conducted on ImageWoof and ImageNette with 10 images per class (IPC = 10). As reported in Table 10, both the prototype-based method and k-means centroids achieve superior performance, slightly outperforming the other strategies. More importantly, when combined with different coreset selection approaches, our method consistently delivers competitive results, demonstrating that it can seamlessly work with various selection mechanisms. This confirms the strong generalization capability and broad applicability of our framework.

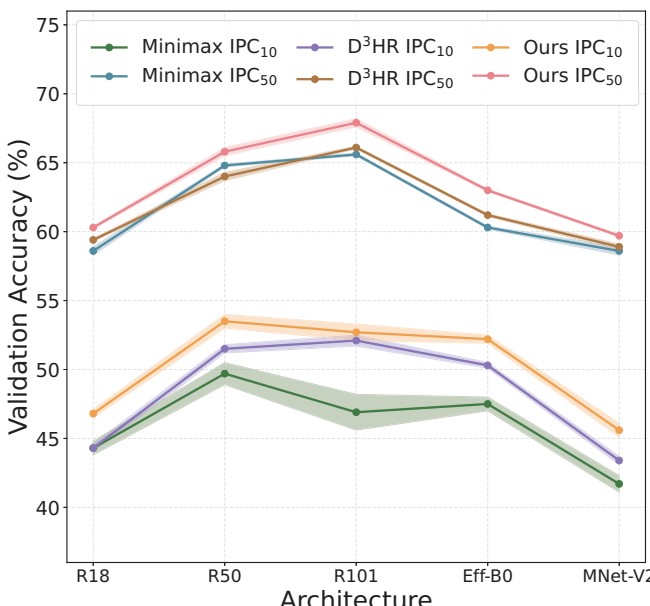

*Figure 9.* Cross-architecture performance evaluated on ImageNet-1K.

*Table 8.* Performance comparison on five ImageNet subsets.

| Distil Alg. | Method | ImageNet-A | ImageNet-B | ImageNet-C | ImageNet-D | ImageNet-E |
|---|---|---|---|---|---|---|
| DC | Pixel | $52.3_{\pm0.7}$ | $45.1_{\pm8.3}$ | $40.1_{\pm7.6}$ | $36.1_{\pm0.4}$ | $38.1_{\pm0.4}$ |
| | GLaD | $53.1_{\pm1.4}$ | $50.1_{\pm0.6}$ | $48.9_{\pm1.1}$ | $38.9_{\pm1.0}$ | $38.4_{\pm0.7}$ |
| | H-GLaD | $54.1_{\pm1.2}$ | $52.0_{\pm0.8}$ | $49.5_{\pm0.9}$ | $39.5_{\pm1.0}$ | $39.1_{\pm0.7}$ |
| | LM3D | $55.2_{\pm1.0}$ | $51.8_{\pm1.4}$ | $49.9_{\pm1.3}$ | $39.9_{\pm1.3}$ | $39.1_{\pm1.3}$ |
| DM | Pixel | $44.4_{\pm0.5}$ | $52.6_{\pm0.4}$ | $50.6_{\pm0.5}$ | $47.5_{\pm0.7}$ | $35.4_{\pm0.4}$ |
| | GLaD | $52.8_{\pm1.0}$ | $51.3_{\pm0.6}$ | $49.7_{\pm0.4}$ | $36.4_{\pm0.4}$ | $38.6_{\pm0.7}$ |
| | H-GLaD | $55.1_{\pm0.5}$ | $54.2_{\pm0.5}$ | $50.8_{\pm0.4}$ | $38.7_{\pm0.4}$ | $39.4_{\pm0.7}$ |
| | LM3D | $57.0_{\pm1.3}$ | $52.3_{\pm1.3}$ | $48.2_{\pm4.9}$ | $39.5_{\pm1.5}$ | $39.4_{\pm1.8}$ |
| – | **Ours** | $\mathbf{64.5}_{\pm1.1}$ | $\mathbf{65.8}_{\pm1.0}$ | $\mathbf{60.1}_{\pm1.3}$ | $\mathbf{47.6}_{\pm0.9}$ | $\mathbf{51.9}_{\pm1.2}$ |

# J. More Implementation Details

In this section, we provide detailed implementation details of our prototype-guided VAR framework.

## J.1. Backbone and Pretrained Models

We adopt the pre-trained VAR-d30 model (Tian et al., 2024) as the generative backbone. The model operates at a resolution of 256×256 and follows a multi-stage autoregressive generation scheme with progressively increasing spatial resolutions.

Specifically, the VAR model consists of 10 autoregressive stages, corresponding to patch resolutions (1, 2, 3, 4, 5, 6, 8, 10, 13, 16), where each stage predicts discrete VQ-VAE tokens that are later decoded into images. The VQ-VAE uses a codebook size of 4096 with an embedding dimension of 32.

All VAR and VQ-VAE parameters are frozen during sampling, and no fine-tuning is performed.

## J.2. Prototype Feature Extraction

To enable semantic guidance during generation, we extract class-specific prototype features using a pre-trained **ResNet-50** model.

*Table 9.* Performance comparison on ImageNet-100.

| IPC (Ratio) | Test Model | Random | Herding | IDC-1 | Minimax | Ours |
|---|---|---|---|---|---|---|
| | ConvNet-6 | $17.0_{\pm 0.3}$ | $17.2_{\pm 0.3}$ | $24.3_{\pm 0.5}$ | $22.3_{\pm 0.5}$ | $24.2_{\pm 0.8}$ |
| 10 (0.8%) | ResNetAP-10 | $19.1_{\pm 0.4}$ | $19.8_{\pm 0.3}$ | $25.7_{\pm 0.1}$ | $24.8_{\pm 0.2}$ | $25.9_{\pm 0.3}$ |
| | ResNet-18 | $17.5_{\pm 0.5}$ | $16.1_{\pm 0.2}$ | $25.1_{\pm 0.2}$ | $22.5_{\pm 0.3}$ | $24.7_{\pm 0.6}$ |
| | ConvNet-6 | $24.8_{\pm 0.2}$ | $24.3_{\pm 0.4}$ | $28.8_{\pm 0.3}$ | $29.3_{\pm 0.4}$ | $31.5_{\pm 0.5}$ |
| 20 (1.6%) | ResNetAP-10 | $26.7_{\pm 0.5}$ | $27.6_{\pm 0.1}$ | $29.9_{\pm 0.2}$ | $32.3_{\pm 0.3}$ | $34.1_{\pm 0.3}$ |
| | ResNet-18 | $25.5_{\pm 0.3}$ | $24.7_{\pm 0.1}$ | $30.2_{\pm 0.2}$ | $31.2_{\pm 0.1}$ | $33.6_{\pm 0.6}$ |

*Table 10.* Generalization analysis across different coreset selection algorithms.

| Coreset Selection Algorithms | ImageWoof | ImageNette |
|---|---|---|
| Random | $41.8_{\pm 1.5}$ | $62.2_{\pm 1.0}$ |
| DBSCAN | $43.6_{\pm 1.7}$ | $63.6_{\pm 1.8}$ |
| $k$-Means | $46.5_{\pm 1.6}$ | $68.2_{\pm 1.1}$ |
| Prototypep Selection | $47.2_{\pm 1.3}$ | $68.0_{\pm 0.9}$ |

**Prototype construction.** For each class, we collect images from a class-specific prototype directory. Representative prototypes are selected using a medoid-based strategy:

**Step 1:** Extract global ResNet-50 features (2048-d).

**Step 2:** Compute the feature centroid.

**Step 3:** Select the image closest to the centroid as the prototype.

**Step 4:** Optionally, additional prototypes are selected using a farthest-first strategy to improve diversity.

**Stage-wise prototype features.** For each selected prototype, we compute stage-specific features to align with the multi-scale nature of VAR. At autoregressive stage $s$ with patch number $p_s$, the prototype image is resized to

$$\max\left(16, \left\lfloor 256 \cdot \frac{p_s}{p_{\max}} \right\rfloor\right), \tag{38}$$

followed by standard ResNet preprocessing and feature extraction. This results in a list of 2048-dimensional prototype features for each stage.

### J.3. Prototype-Guided Autoregressive Sampling

We modify the original classifier-free guidance (CFG) sampler in VAR by introducing **prototype-aware semantic biasing** at intermediate stages.

**Guidance interval.** Prototype guidance is applied only at intermediate stages (stages [0.2, 0.8]), motivated by the observation that semantic structures are primarily formed during mid-generation (Yu et al., 2023). Early and late stages are left largely unaffected.

**Expected Embedding Approximation.** At VAR stage $s$, the model produces logits $\ell^{(s)} \in \mathbb{R}^{B \times L_s \times V}$ with corresponding probabilities $\Pi^{(s)} = \text{softmax}(\ell^{(s)})$. For batch index $b$ and token position $i$, we compute the expected semantic embedding as

$$\bar{h}_{b,i}^{(s)} = \Pi_{b,i,:}^{(s)} E = \mathbb{E}_{v \sim \Pi_{b,i,:}^{(s)}}[E_v] \in \mathbb{R}^C, \tag{39}$$

where $E_v$ denotes the embedding of token $v$ in the vocabulary. Aggregating $\{\bar{h}_{b,i}^{(s)}\}_{i=1}^{L_s}$ yields a feature map $\hat{H}_b^{(s)} \in \mathbb{R}^{C \times h_s \times w_s}$, which is decoded by the VQ-VAE proxy into a tentative image $\hat{x}_b^{(s)}$. We further extract its semantic representation $f(\hat{x}_b^{(s)}) \in \mathbb{R}^d$ using a ResNet-50 encoder.

**Semantic Similarity and Logit Biasing.** Given the class prototype $p_{c,s}$ at stage $s$, we compute the cosine similarity

$$\text{sim}_b^{(s)} = \cos\big(f(\hat{x}_b^{(s)}), p_{c,s}\big), \tag{40}$$

which measures the agreement between the current image trajectory and the prototype manifold. This similarity is modulated by a stage-aware Gaussian schedule

$$\kappa(s) = \exp\left(-\frac{(\rho_s - 0.5)^2}{2\sigma^2}\right), \qquad \rho_s = s/S. \tag{41}$$

The resulting guidance signal is incorporated into the logit refinement process by jointly considering image–prototype similarity and token–semantic alignment, and is added to the original VAR logits prior to sampling.

### J.4. Guidance Hyper-parameters

The prototype guidance hyper-parameters are summarized in Table 11.

*Table 11.* Prototype guidance settings.

| Parameter | Value |
|---|---|
| Guidance strength $\lambda$ | 0.015 |
| Gaussian std $\sigma$ | 0.15 |
| Effective guidance interval | $[0.2, 0.8]$ |
| Similarity metric | Cosine |

These values are fixed across all experiments unless otherwise specified.

### J.5. Candidate Pool Generation

For each class, we generate a pool of synthetic images using the guided VAR sampler.

- Pool size: $5 \times$ IPC (up to 1000 images per class)

- Batch size: 20

- CFG scale: 5

- Sampling strategy: top-$k = 900$, top-$p = 0.96$

Each generated image is encoded using ResNet-50 to obtain a 2048-dimensional feature vector.

### J.6. Teacher Filtering and Diversity Selection

To improve sample quality and diversity, we adopt a two-stage selection strategy.

**Teacher filtering.** A pre-trained ImageNet classifier (e.g., ResNet-18) is used to filter the candidate pool:

- Correctly classified samples are prioritized.

- Samples are ranked by prediction confidence.

- If insufficient correct samples are available, high-confidence incorrect samples are added.

### J.7. Computational Cost

All experiments are conducted on NVIDIA RTX 4090 or RTX A6000 GPUs. Prototype-guided sampling is performed on a single NVIDIA RTX 4090 GPU. For ImageNet-1K, generating a pool of 1,000 images per class can be completed within a few dozen hours on **a single 4090 GPU**.

For evaluation, experiments on datasets with a small number of classes (e.g., 10 classes) are conducted on an NVIDIA RTX 4090 GPU. Due to the memory constraints of the RTX 4090, evaluations on ImageNet-1K are carried out on NVIDIA RTX A6000 GPUs.

### J.8. Validation Protocol.

For validation, we follow the configurations reported in Sun et al. (2024) for fair comparison, as summarized in Table 12. We evaluate the model by training a ResNet-18 network for 300 epochs, which is used as the teacher. We adopt the AdamW optimizer with a learning rate of 0.001 and a weight decay of 0.01. For other baselines, results are either reported from the original papers or reproduced using the authors' official code under the same evaluation settings.

*Table 12.* Hyper-parameter settings used for validation.

| Parameter | Value |
|---|---|
| Optimizer | AdamW |
| Learning Rate | 0.001 |
| Weight Decay | 0.01 |
| Batch Size | 128 |
| Data Augmentation | RandomResizedCrop + RandomHorizontalFlip |
| Epochs | 300 |

## K. Detailed Analysis of the ProtoVAR Framework

In this section, we provide further analysis and ablation studies regarding the design of ProtoVAR, specifically addressing the role of target-guided filtering and the impact of model capacity.

### K.1. Effectiveness of Target-Guided Filtering

**Efficiency-unlocked Capability.** A key concern is whether the use of an external teacher classifier for filtering creates an unfair comparison. We argue that target-guided filtering is not merely an extra trick, but a practical capability uniquely unlocked by VAR's efficiency. Filtering requires generating a large candidate pool (e.g., a $5\times$ pool for ImageNet-1K at IPC 50 involves generating $\sim$250k images). For diffusion models (e.g., DiT), generating such a pool via sequential denoising is computationally prohibitive. In contrast, VAR's parallel prediction is faster than DiT (see Table 13), making large-scale filtering a viable and cost-effective component of the distillation pipeline.

*Table 13.* Comparison of inference efficiency.

| Method | Time/Img (s) |
|---|---|
| DDIM | 1.2 |
| DiT-XL/2 | 0.8 |
| ProtoVAR | **0.08** |

**Cross-method Comparison with Filtering.** As demonstrated in Section 4.2, ProtoVAR achieves highly competitive performance even without the teacher-guided filtering stage. To further analyze the impact of this filtering mechanism across different generative frameworks, we equip all baseline methods with the same teacher classifier. As shown in Table 14, while all methods benefit from quality control, ProtoVAR maintains a significant performance lead, further validating the superiority of our prototype-guided generation.

*Table 14.* Comparison with baselines when all methods are equipped with the same target-guided filter.

| Method (+Filter) | Woof | | Nette | |
| --- | --- | --- | --- | --- |
| | IPC 10 | IPC 50 | IPC 10 | IPC 50 |
| Minimax | $41.7_{\pm1.5}(\uparrow 1.6)$ | $68.9_{\pm2.1}(\uparrow 1.9)$ | $62.9_{\pm0.9}(\uparrow 1.5)$ | $84.9_{\pm1.3}(\uparrow 0.8)$ |
| D$^4$M | $39.6_{\pm1.7}(\uparrow 2.1)$ | $67.0_{\pm1.9}(\uparrow 1.3)$ | $60.2_{\pm1.8}(\uparrow 0.9)$ | $83.6_{\pm1.5}(\uparrow 1.1)$ |
| D$^3$HR | $44.3_{\pm1.1}(\uparrow 3.2)$ | $71.6_{\pm1.2}(\uparrow 1.5)$ | $64.2_{\pm1.0}(\uparrow 1.4)$ | $85.6_{\pm1.3}(\uparrow 1.0)$ |
| ProtoVAR (Ours) | $\mathbf{47.2_{\pm1.3}}(\uparrow \mathbf{4.0})$ | $\mathbf{72.0_{\pm0.7}}(\uparrow \mathbf{2.5})$ | $\mathbf{68.0_{\pm0.9}}(\uparrow \mathbf{2.7})$ | $\mathbf{85.8_{\pm0.8}}(\uparrow \mathbf{3.7})$ |

## K.2. Sensitivity to Model Capacity

We evaluate the impact of model capacity for both the prototype extractor and the teacher filter.

**Prototype Extractor.**   As shown in Table 15, performance remains remarkably stable across various backbones. Notably, even a randomly initialized ResNet50 provides sufficient guidance for capturing data distribution via cosine similarity.

*Table 15.* Ablation on Prototype Extractor backbone capacity.

| Extractor | Woof | | Nette | |
| --- | --- | --- | --- | --- |
| | IPC 10 | IPC 50 | IPC 10 | IPC 50 |
| Random ResNet50 | $46.3_{\pm1.2}$ | $71.3_{\pm1.5}$ | $66.9_{\pm1.4}$ | $85.2_{\pm1.6}$ |
| ConvNet | $46.5_{\pm1.1}$ | $71.6_{\pm1.3}$ | $67.8_{\pm1.2}$ | $85.2_{\pm1.4}$ |
| MobileNetV2 | $46.9_{\pm0.9}$ | $71.8_{\pm1.1}$ | $68.1_{\pm1.0}$ | $85.5_{\pm1.1}$ |
| ResNet50 (Ours) | $47.2_{\pm1.3}$ | $72.0_{\pm0.7}$ | $68.0_{\pm0.9}$ | $85.8_{\pm0.8}$ |
| ResNet101 | $47.4_{\pm1.1}$ | $71.9_{\pm0.8}$ | $68.2_{\pm1.1}$ | $85.6_{\pm0.9}$ |

**Teacher Filter.**   Table 16 demonstrates that while pre-trained teachers improve filtering accuracy, the framework is robust to smaller models. ResNet18 offers the optimal trade-off between efficiency and selection quality.

*Table 16.* Ablation on Teacher Filter backbone capacity.

| Teacher Filter | Woof | | Nette | |
| --- | --- | --- | --- | --- |
| | IPC 10 | IPC 50 | IPC 10 | IPC 50 |
| Random ResNet18 | $45.1_{\pm1.1}$ | $70.6_{\pm1.3}$ | $66.3_{\pm1.3}$ | $84.2_{\pm1.5}$ |
| ConvNet | $45.9_{\pm0.9}$ | $71.2_{\pm1.0}$ | $66.8_{\pm1.0}$ | $84.7_{\pm1.2}$ |
| MobileNetV2 | $46.3_{\pm0.8}$ | $71.4_{\pm0.9}$ | $67.1_{\pm0.9}$ | $85.2_{\pm1.0}$ |
| ResNet18 (Ours) | $47.2_{\pm1.3}$ | $72.0_{\pm0.7}$ | $68.0_{\pm0.9}$ | $85.8_{\pm0.8}$ |
| ResNet50 | $47.3_{\pm1.1}$ | $72.2_{\pm0.8}$ | $68.0_{\pm1.0}$ | $85.9_{\pm0.7}$ |

## K.3. Generalization Across Evaluator Architectures

To ensure ProtoVAR does not overfit to the teacher's architecture, we conducted cross-architecture evaluations. The results in Table 17 confirm that the distilled dataset achieves strong performance across diverse evaluators, regardless of the teacher model used for filtering.

# L. Effect of Guidance Strength $\lambda$

Figure 10 illustrates the effect of the guidance strength $\lambda$ on performance. We observe that within a moderate range of values, the performance remains stable. However, as $\lambda$ increases further, excessively strong prototype guidance reduces sample diversity, leading to performance degradation. This phenomenon is further corroborated by the t-SNE visualizations in

Table 17. Cross-architecture evaluation on ImageWoof (IPC 10) with various teacher filters.

| Teacher / Evaluator | ResNet18 | ConvNet | MobileNet | ViT |
|---|---|---|---|---|
| ConvNet | $45.9_{\pm1.2}$ | $39.4_{\pm1.3}$ | $45.1_{\pm1.2}$ | $44.9_{\pm1.1}$ |
| MobileNet | $46.3_{\pm1.3}$ | $40.2_{\pm1.2}$ | $45.6_{\pm1.0}$ | $\mathbf{45.5}_{\pm1.0}$ |
| ResNet18 (Ours) | $\mathbf{47.2}_{\pm1.3}$ | $\mathbf{40.6}_{\pm1.1}$ | $\mathbf{46.3}_{\pm0.9}$ | $45.1_{\pm0.8}$ |

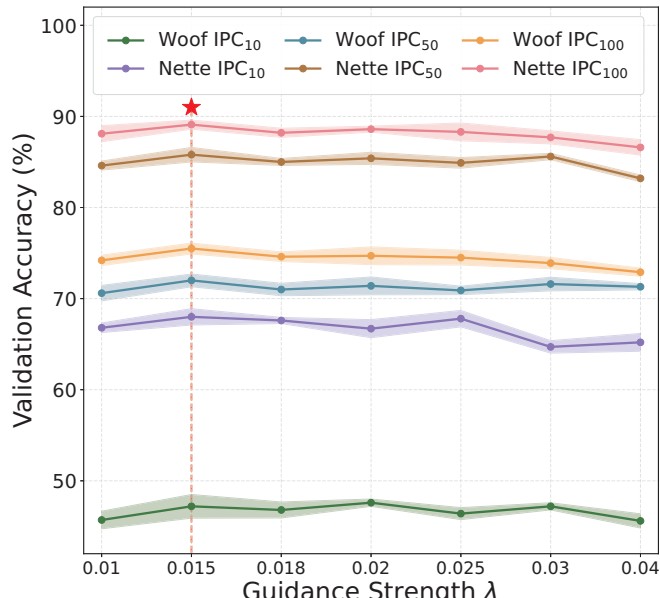

Figure 10. Sensitivity to the guidance strength $\lambda$.

Figure 4, where overly strong guidance causes the generated samples to cluster tightly around their corresponding prototypes, resulting in over-concentrated feature distributions. Based on these observations, we set the guidance strength $\lambda$ to 0.015.

## M. Effect of Prototype Count and Pool Size

The number of prototypes directly affects the diversity introduced by prototype-guided generation. Figure 11a analyzes its impact on performance. When the prototype count is extremely small, the provided supervision is limited; nevertheless, the performance already surpasses VAR, which is consistent with the single-prototype results shown in Figure 4. As the number of prototypes increases within a moderate range, the model achieves stable performance improvements. However, under the IPC 10 setting, further increasing the prototype count leads to performance degradation. We attribute this to the instability introduced by excessive prototypes, which impose overly restrictive guidance and hinder effective sample generation under limited IPC budgets.

The confidence pool size determines the number of candidate samples participating in the final IPC selection. Figure 11b presents its influence on performance. We observe that increasing the pool size within a reasonable range consistently improves performance. In particular, under IPC 10, a larger pool size yields better results, as it provides richer candidate samples and captures more informative variations. Considering both performance and storage efficiency, we set the prototype count to 5 and adopt a confidence pool size of 5×IPC in all experiments.

## N. Robustness and Generalization Analysis

A key advantage of ProtoVAR is that it requires no dataset-specific tuning. To verify this, we evaluate the robustness of our default settings ($\lambda = 0.015$, $\sigma = 0.15$) across diverse datasets.

The choice of the scheduling parameter $\sigma$ is grounded in the intrinsic generative dynamics of VAR. During the early stages

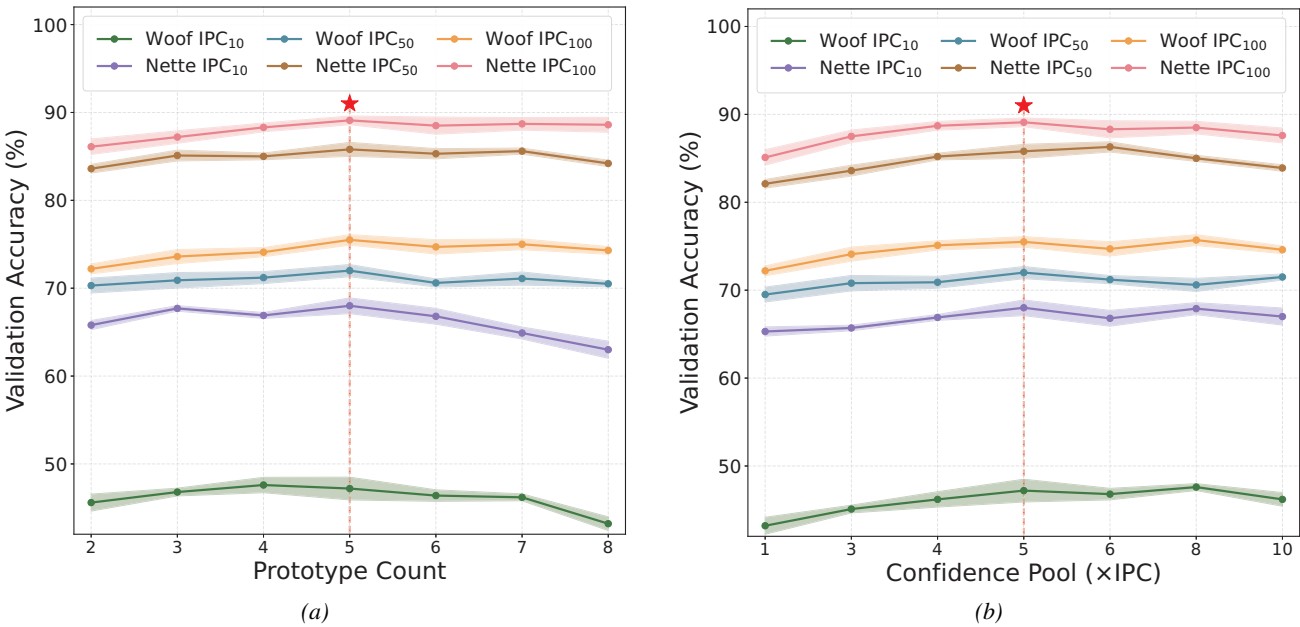

*Figure 11.* Hyperparameter Analysis. (a) Sensitivity to the prototype count. (b) Sensitivity to the confidence pool size (×IPC). All results in (a) and (b) are obtained using ResNet-18. The dashed line indicates the value adopted in this work.

*Table 18.* Cross-dataset generalization with fixed hyperparameters ($\lambda = 0.015, \sigma = 0.15$). All results are reported under IPC 10.

| Dataset | Resolution | RDED | $CaO_2$ | ProtoVAR (Ours) |
|---|---|---|---|---|
| CIFAR-10 | $32 \times 32$ | $37.1_{\pm 0.3}$ | $39.0_{\pm 1.5}$ | $\mathbf{44.6}_{\pm 0.9}$ |
| CIFAR-100 | $32 \times 32$ | $42.6_{\pm 0.2}$ | $48.9_{\pm 0.3}$ | $\mathbf{53.2}_{\pm 0.3}$ |
| Tiny-ImageNet | $64 \times 64$ | $41.9_{\pm 0.2}$ | $46.3_{\pm 0.5}$ | $\mathbf{51.6}_{\pm 0.2}$ |
| FooD-101 | $512 \times 512$ | $22.8_{\pm 1.1}$ | — | $\mathbf{28.3}_{\pm 0.8}$ |

of generation, strong guidance can disrupt the global geometric layout, while in the late stages, it may suppress fine-grained texture diversity. By setting $\sigma = 0.15$, we target the middle stages of the "coarse-to-fine" process to consolidate core semantics, which serves as a universal optimum across different data distributions.

Furthermore, we observe that performance remains stable across a broad range of guidance strengths $\lambda \in [0.01, 0.02]$. As shown in Table 18, using the same default hyperparameters, ProtoVAR consistently outperforms state-of-the-art methods on various datasets, including CIFAR and Food-101. These results demonstrate that our guidance mechanism is robust to changes in image resolution and category scales, eliminating the need for extensive manual tuning.

## O. Additional Visualization Comparisons

We present a comparison between the samples generated by VAR and our proposed ProtoVAR on the ImageNet subset in Figures 12 to 21. In most cases, VAR produces realistic images that roughly capture the characteristics of the original dataset. However, due to the lack of explicit guidance at the generation stage, the diversity of synthesized samples is limited. For example, most generated images of the "Golden Retriever" class mainly focus on the head region, reflecting a biased coverage of visual patterns. In contrast, our prototype-guided VAR generates samples that are not only representative but also exhibit greater diversity, which is further supported by the performance gains shown in Table 1. Meanwhile, there remain some challenging cases that suggest room for improvement. We regard these limitations as promising directions for future work, both for VAR-based models and for generative dataset distillation methods.

VAR ProtoVAR

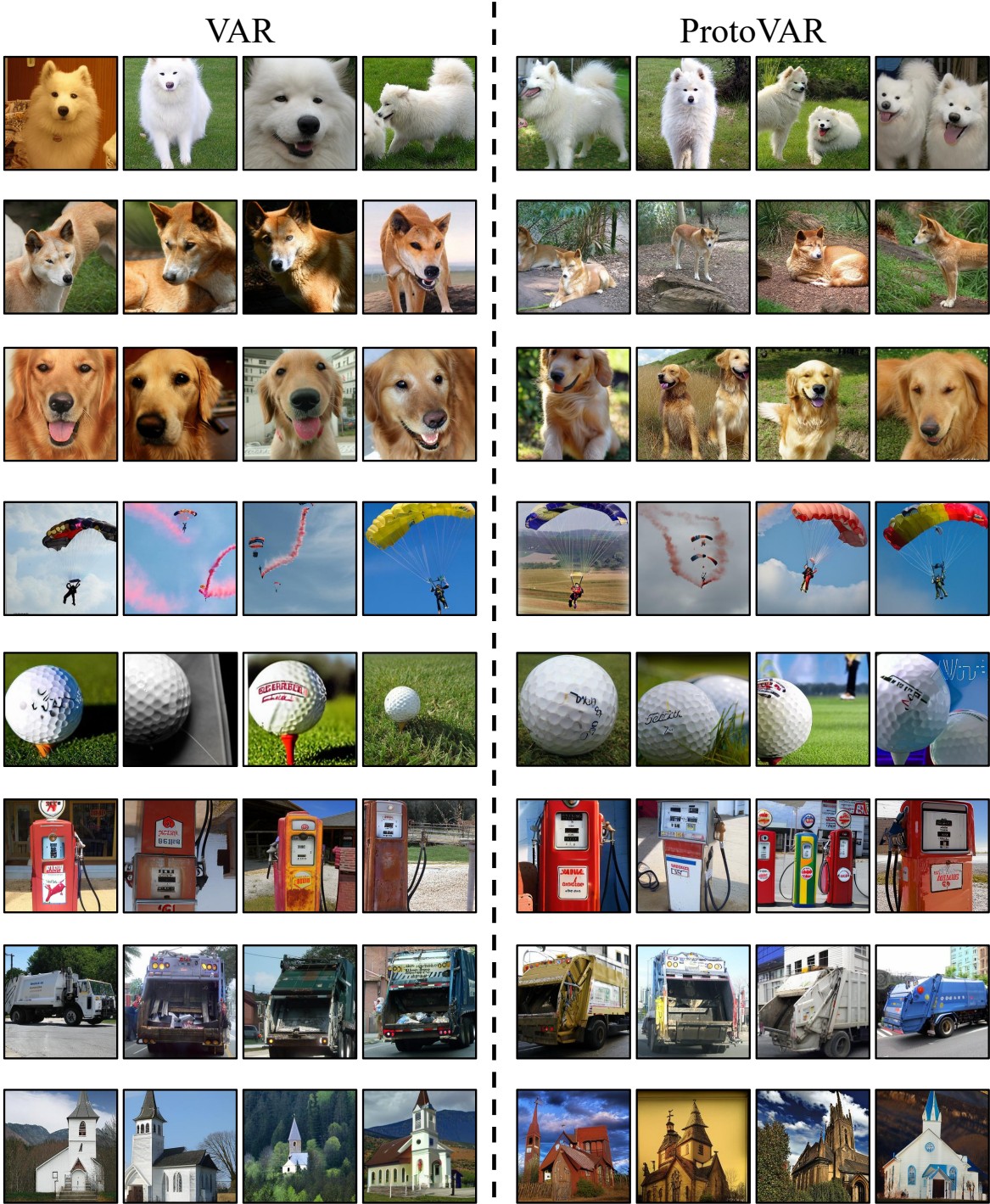

*Figure 12.* Comparison of samples generated by VAR and our ProtoVAR on the ImageNet subset.

VAR · ProtoVAR

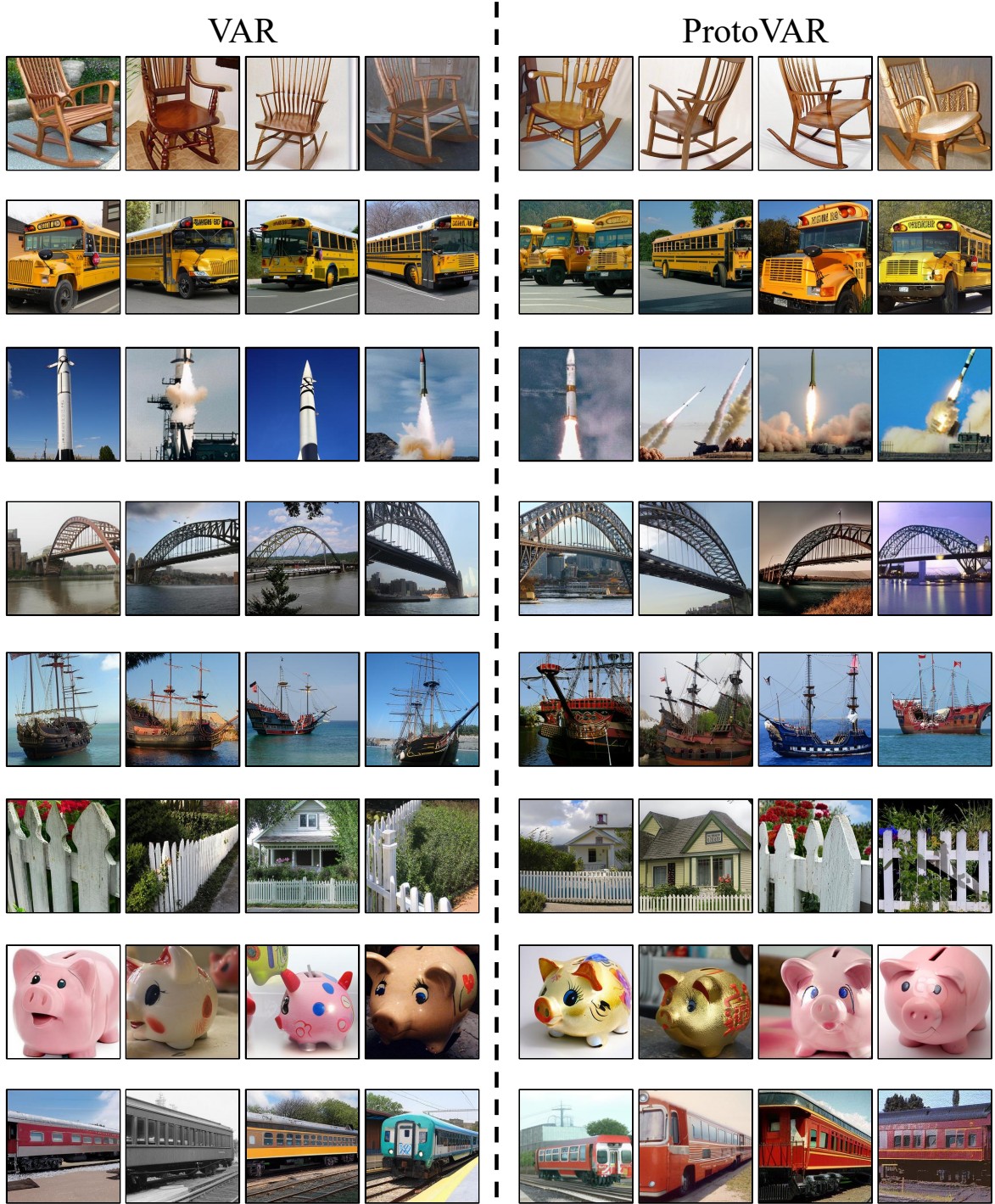

*Figure 13.* Comparison of samples generated by VAR and our ProtoVAR on the ImageNet subset.

VAR                                    ProtoVAR

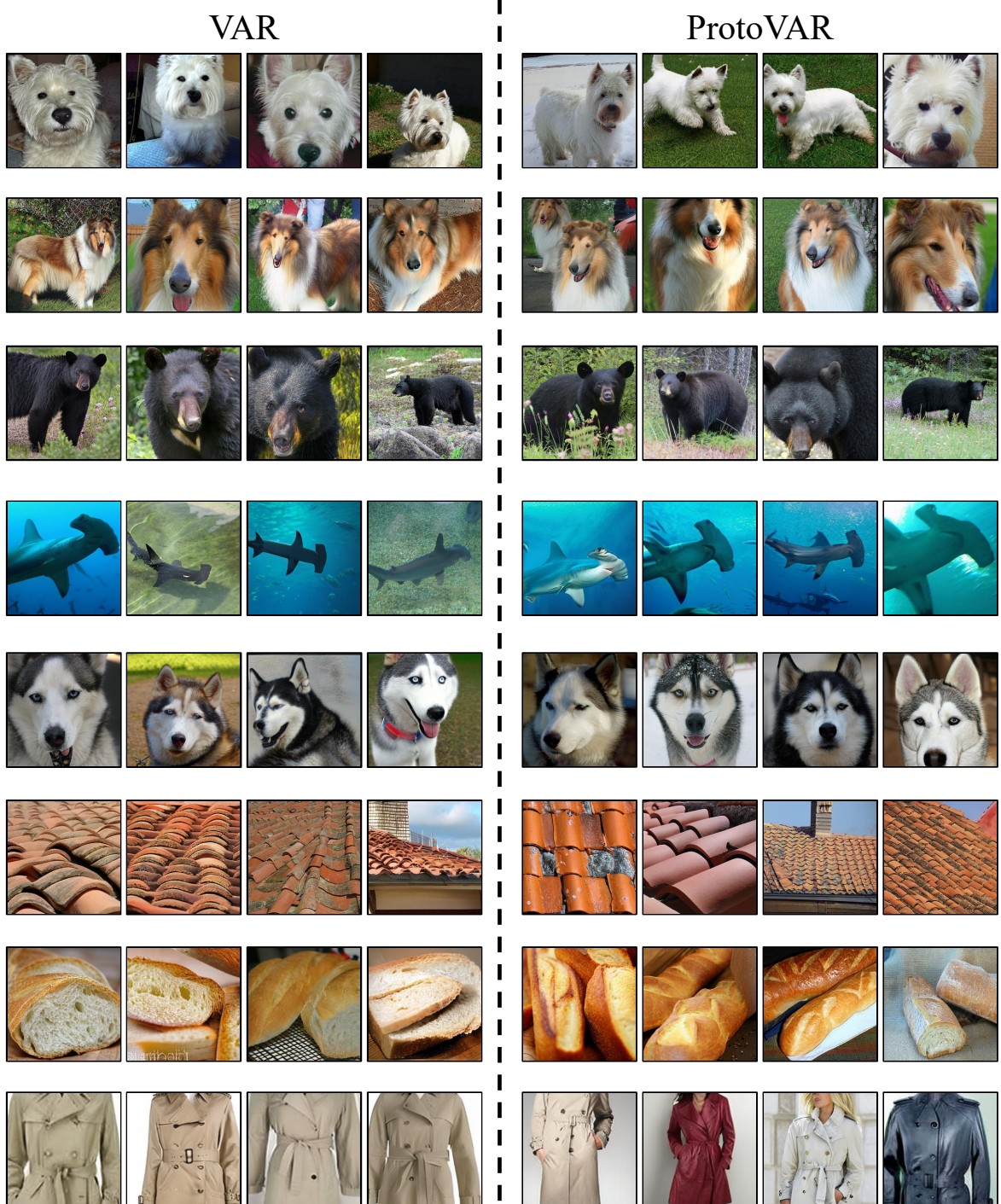

*Figure 14.* Comparison of samples generated by VAR and our ProtoVAR on the ImageNet subset.

## VAR ProtoVAR

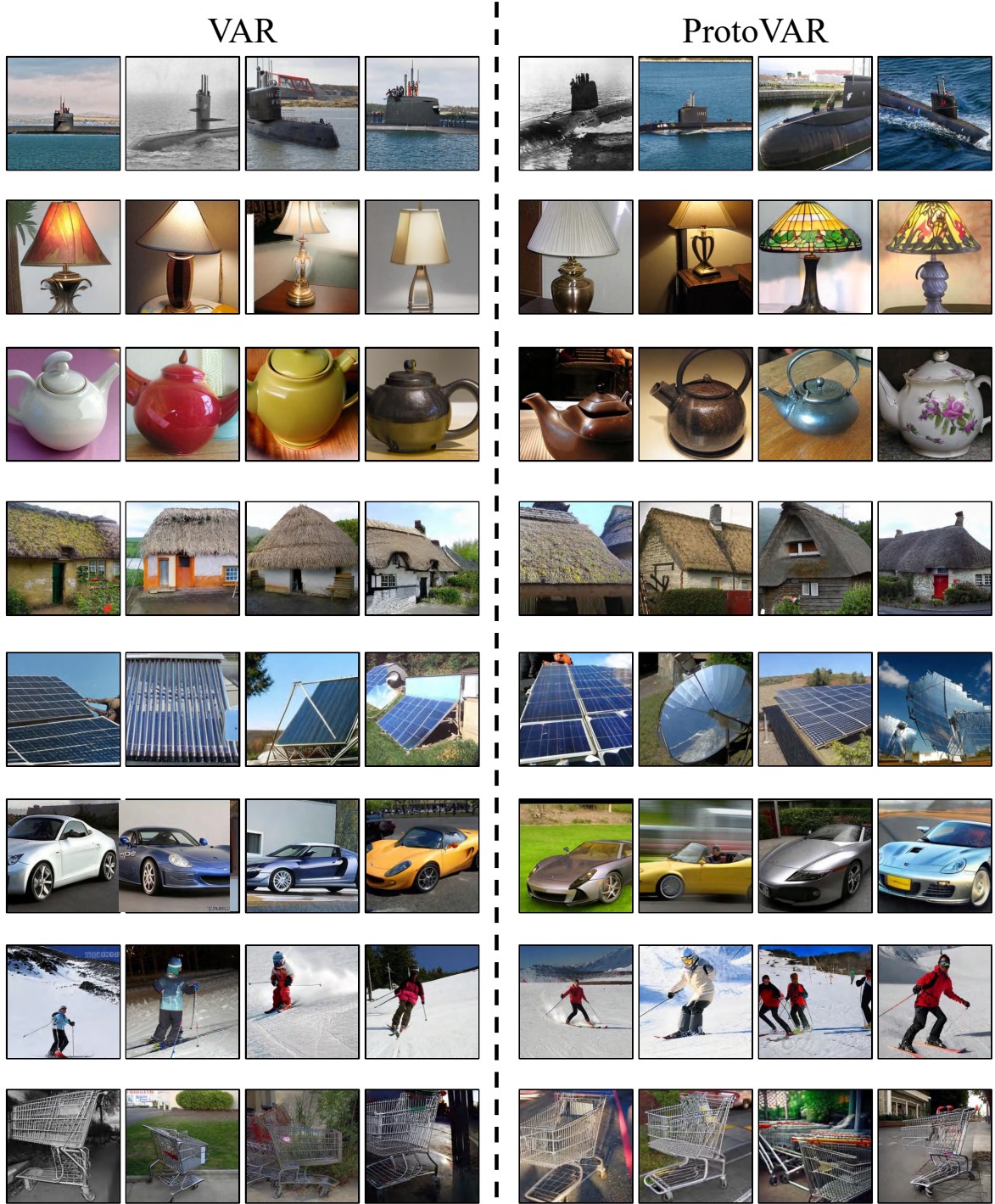

*Figure 15.* Comparison of samples generated by VAR and our ProtoVAR on the ImageNet subset.

VAR ProtoVAR

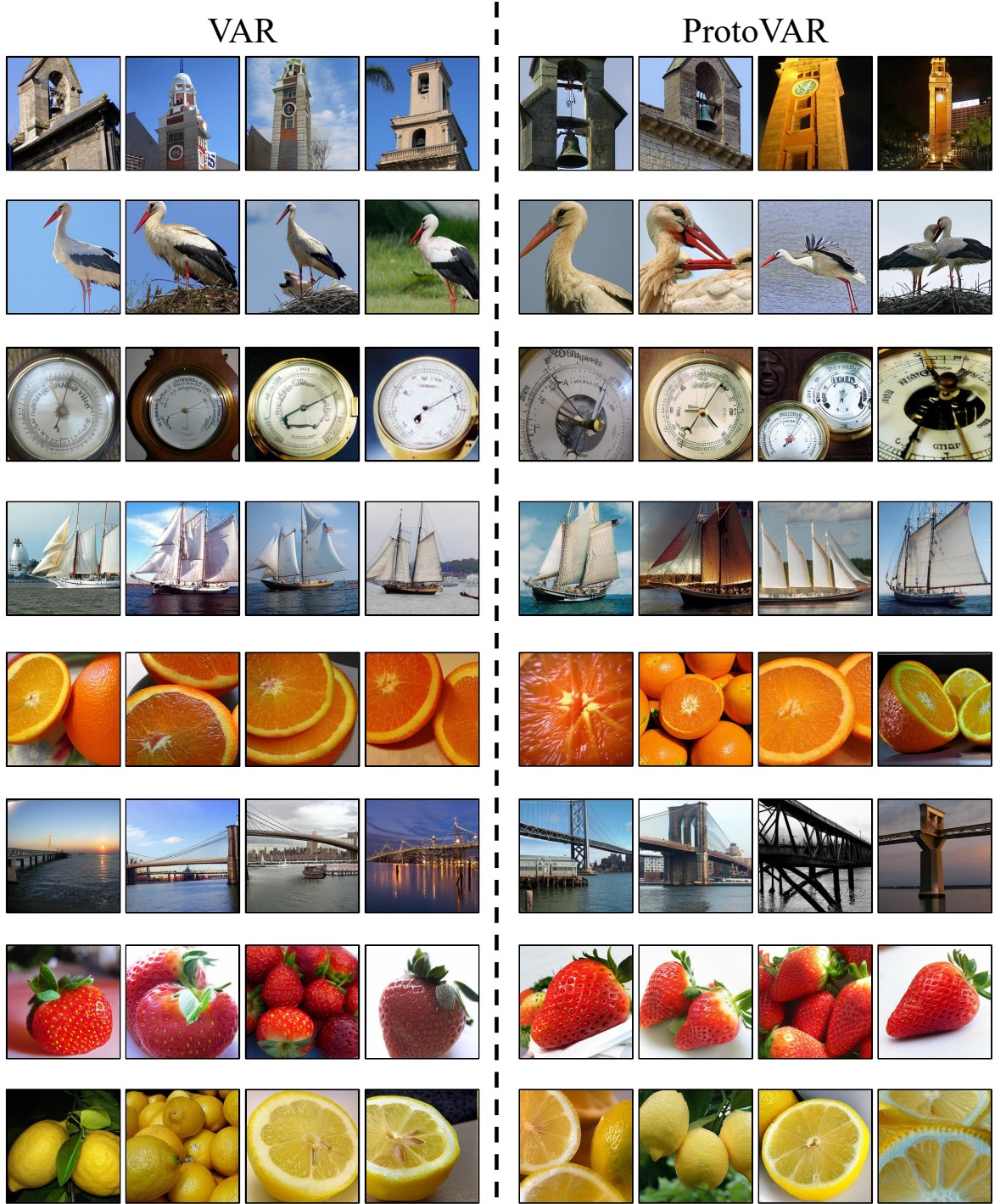

*Figure 16.* Comparison of samples generated by VAR and our ProtoVAR on the ImageNet subset.

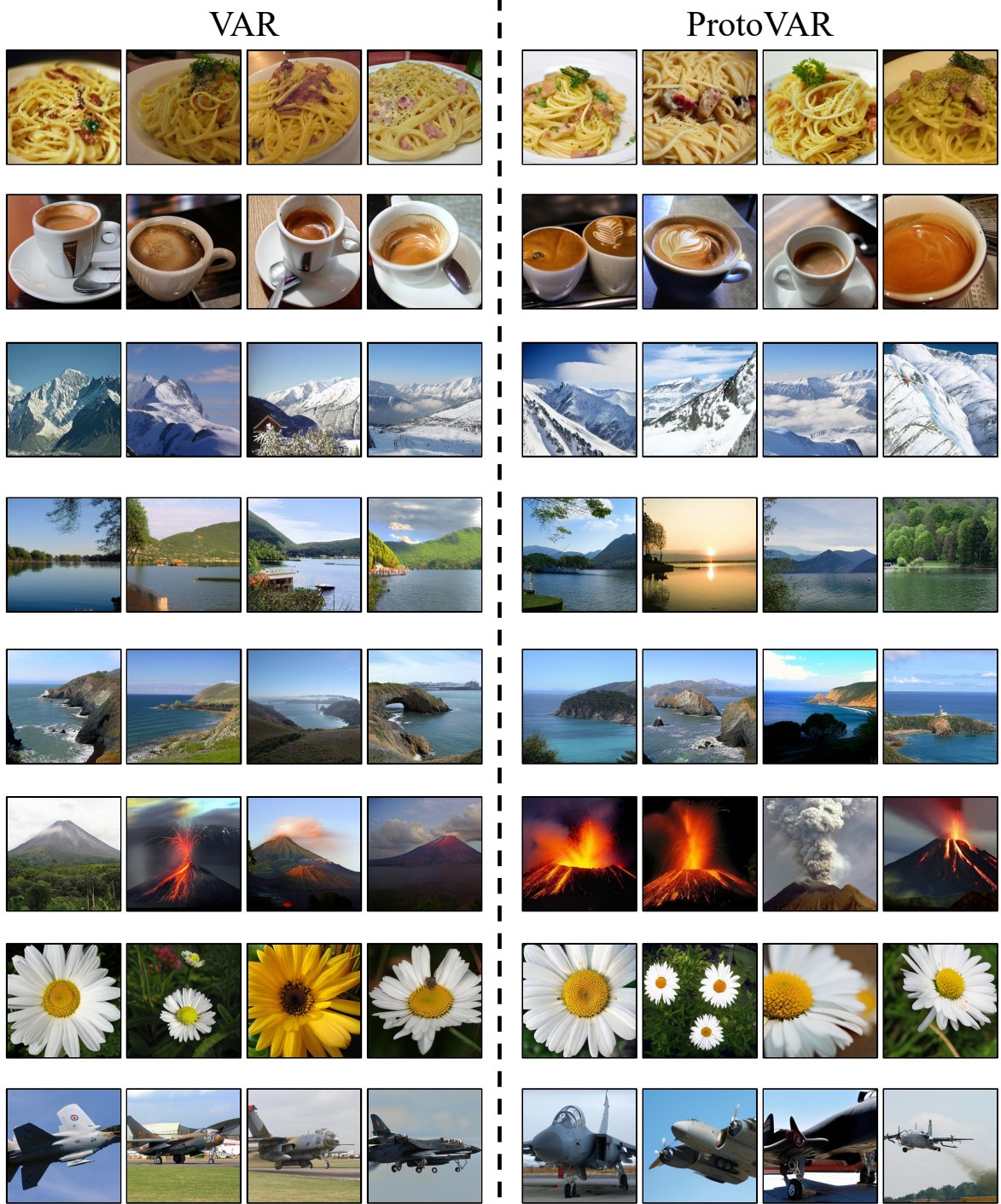

*Figure 17.* Comparison of samples generated by VAR and our ProtoVAR on the ImageNet subset.

# VAR ProtoVAR

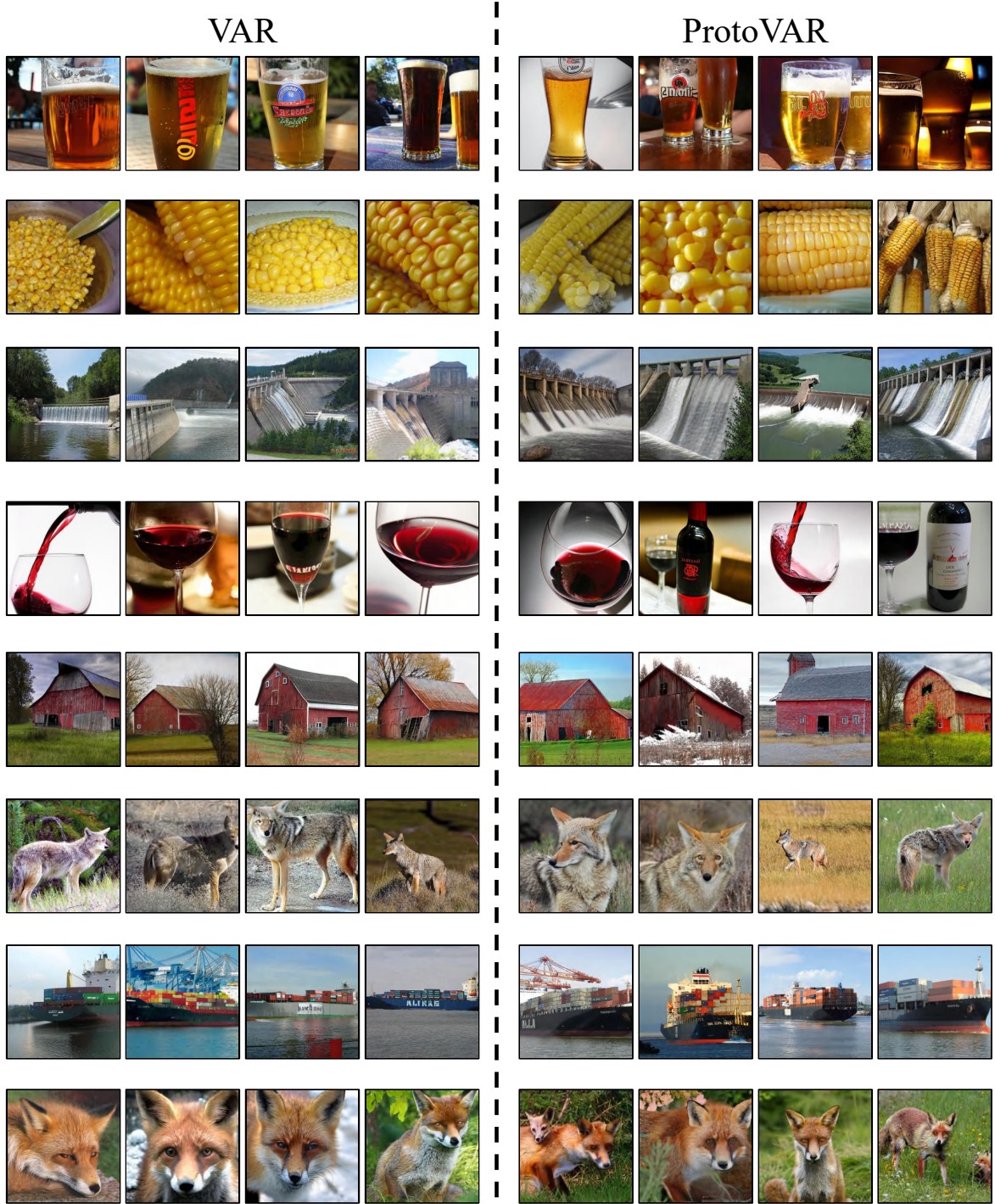

*Figure 18.* Comparison of samples generated by VAR and our ProtoVAR on the ImageNet subset.

VAR ProtoVAR

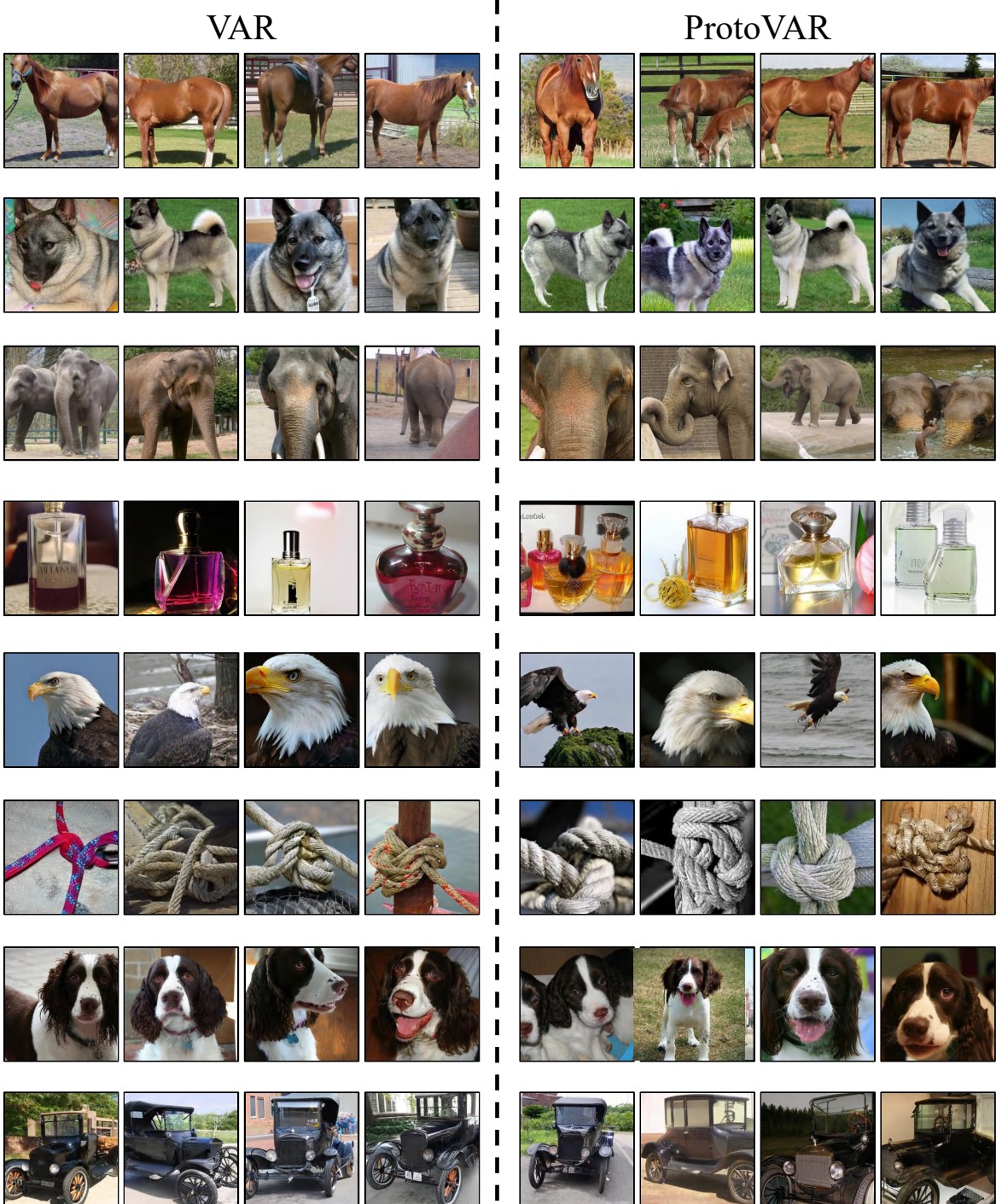

*Figure 19.* Comparison of samples generated by VAR and our ProtoVAR on the ImageNet subset.

VAR ProtoVAR

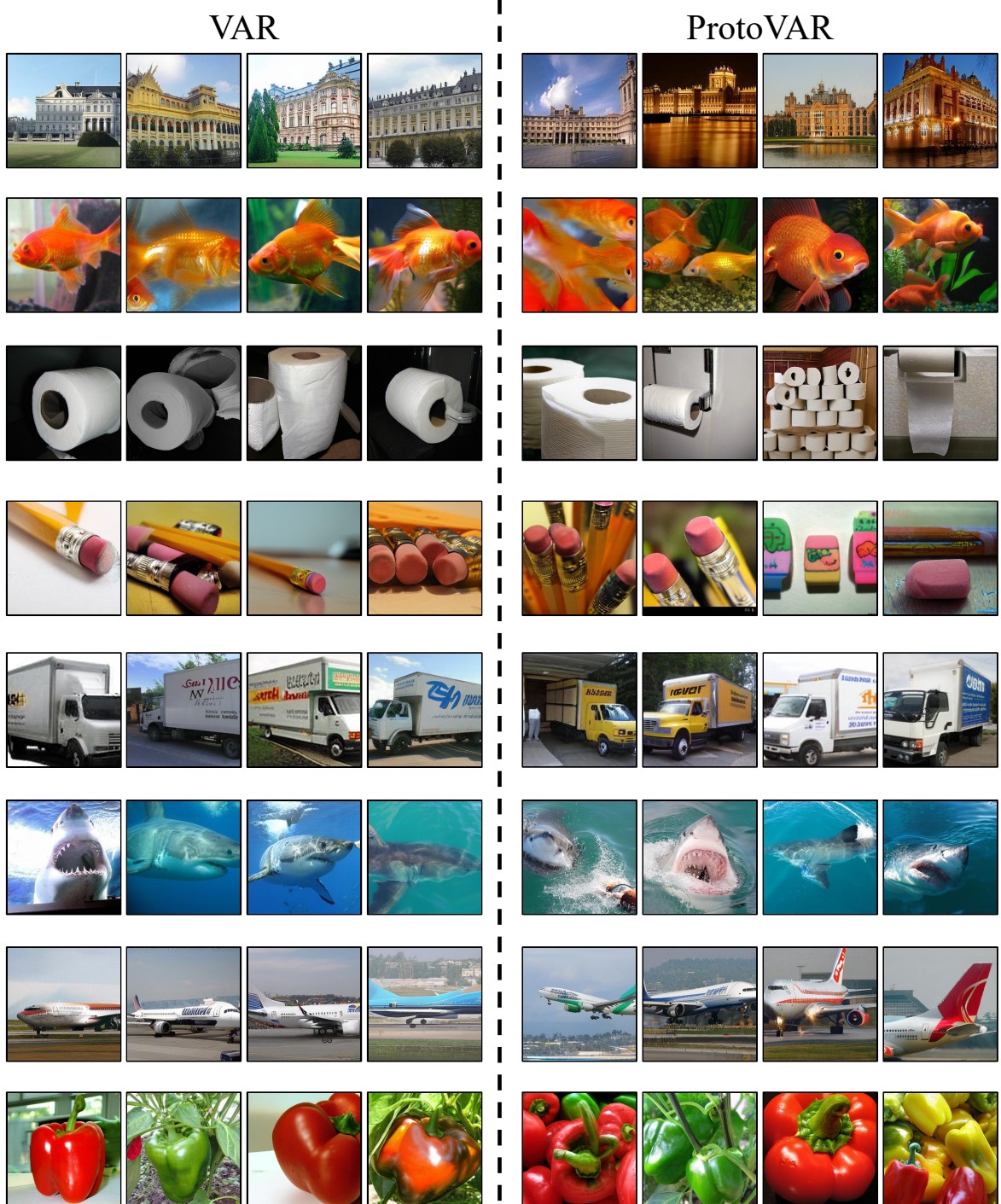

*Figure 20.* Comparison of samples generated by VAR and our ProtoVAR on the ImageNet subset.

VAR | ProtoVAR

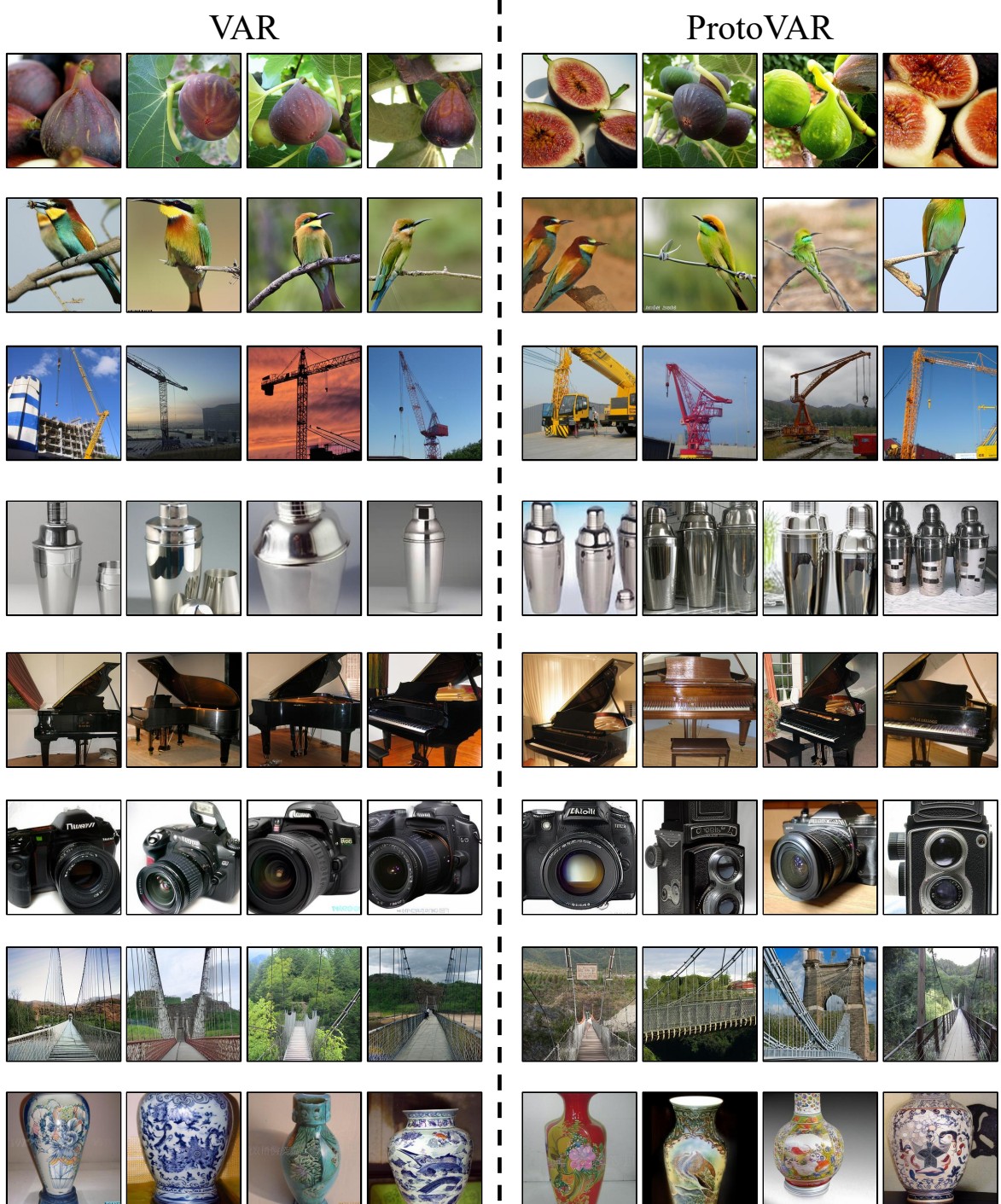

*Figure 21.* Comparison of samples generated by VAR and our ProtoVAR on the ImageNet subset.

