# OpenReview forum: "ProtoVAR: Efficient Dataset Distillation via Prototype-Guided Visual Autoregressive Modeling"
_ICML.cc/2026/Conference — ICML 2026 regular_

### Official Review · Reviewer_UJL9 · 2026-03-10

**Soundness:** 3
**Presentation:** 3
**Significance:** 2
**Originality:** 2
**Overall Recommendation:** 4
**Confidence:** 2

**Summary:**

The paper proposes ProtoVAR, a framework for generative dataset distillation that replaces the standard diffusion-based backbones with VAR modeling. The authors address the limitations of diffusion models by introducing Prototype-Guided Logit Refinement and Target-Guided Filtering. ProtoVAR uses multi-scale class prototypes to steer the autoregressive generation process, achieving high-quality surrogate datasets with superior efficiency.

**Compliance With Llm Reviewing Policy:**

Affirmed.

**Final Justification:**

I thank the authors for their detailed response and the additional experimental results. While I still have reservations regarding the heavy reliance on the inference speed of pre-trained models and believe the 'generate-and-filter' approach involves a degree of structural redundancy that could be further optimized, I acknowledge that these challenges are somewhat inherent to the current state of generative dataset distillation. Considering these field-wide limitations and the performance gains demonstrated, I have decided to raise my rating to 4.

**Key Questions For Authors:**

- The proposed guidance mechanism relies on specific values for strength and scheduling. Are these optimal settings robust across diverse datasets with different distributions or scales without requiring extensive manual tuning?

- While cross-architecture performance is claimed, the primary evaluations focus predominantly on ConvNet backbones. Could the authors provide a more detailed individual performance analysis for Vision Transformer (ViT) student models in the main benchmark settings to verify generalization across non-convolutional architectures?

**Limitations:**

yes

**Strengths And Weaknesses:**

Strengths

- The paper is well written and easy to follow.

- By leveraging a VAR backbone, the proposed method accelerates condensed dataset sampling while achieving state-of-the-art results on ResNet and ImageNet subset benchmarks.

Weaknesses

- To enhance task alignment, the method generates a candidate pool. Although VAR sampling is efficient, the necessity of generating a large volume of data only to discard a significant portion during the filtering stage introduces a structural redundancy in computational resource usage.

- The success of the guidance mechanism relies on the careful tuning of the guidance strength and the Gaussian schedule standard deviation. As shown in the ablation studies, sub-optimal values for these parameters can lead to either a loss of diversity or a collapse in representativeness, requiring manual intervention for different datasets.

- The framework's performance is strictly tied to the availability of a pre-trained VAR model and a ResNet feature extractor. In domains where such pre-trained foundational models are unavailable, the initial cost of preparing these models may negate the efficiency gains of the distillation process.


- While cross-architecture generalization is claimed, the main evaluations (Table 1, Figure 9) focus almost exclusively on ConvNet backbones, and a dedicated, independent analysis of student models like Vision Transformers (ViT) is notably absent from the primary benchmark results.

- The framework's performance is strictly tied to the availability of a pre-trained VAR model and a ResNet feature extractor. In domains where such pre-trained foundational models are unavailable, the initial cost of preparing these models may negate the efficiency gains of the distillation process.

---

> ### Author Rebuttal · Authors · 2026-03-31
>
> Thank you for your detailed comments. Please find the responses below:
>
> **W1.Efficiency of Candidate Pool**
>
> This strategy is a deliberate design to bypass the high computational costs of iterative optimization while ensuring high-quality distillation.
>
> 1.Computational Economy
>
> * Previous methods (matching trajectories/gradients) rely on iterative pixel-level updates that scale poorly.
> * VAR generates tokens in parallel different scales, achieving ~70x faster inference than diffusion models. Given this computational efficiency, generating a larger pool of images takes a fraction of the time and memory compared to optimizing a smaller batch of images.
> * As shown in Fig.1 and response to Reviewer Rndg (W1.2), our method reduces both time and GPU memory usage. Discarding part of the pool is not a waste of resources; rather, it is an efficient shortcut that avoiding hundreds of GPU optimization hours. Furthermore, please refer to Reviewer q9Su (W1), even without the pool, ProtoVAR still achieves performance comparable to or exceeding current SOTAs with maximized efficiency.
>
> 2.Paradigm Shift
>
> The "redundancy" is an efficient generate-and-filter paradigm. Instead of costly pixel-by-pixel optimization of exact IPC over hundreds of hours, VAR's speed allows cheap over-generation. Filtering curates the most informative samples, maximizing knowledge density and performance with lower overall time and memory footprints.
>
> **W2\Q1.Hyper-parameter Robustness**
>
> ProtoVAR requires no dataset-specific tuning. Defaults ($\lambda=0.015$, $\sigma=0.15$) are grounded in the autoregressive generation, ensuring cross-dataset robustness.
>
> The choice of $\sigma$ reflects VAR's intrinsic coarse-to-fine dynamics:
> * Early: Establish geometry; strong guidance disrupts layouts.
> * Late: Resolve textures; guidance suppresses diversity.
> * Middle: Consolidate core semantics.
> Setting $\sigma=0.15$ targets this fundamental "information dynamic" during the middle stages, serving as a universal optimum across datasets.
>
> Unlike diffusion models' noisy states, VAR provides deterministic intermediate structures. $\lambda$ scales dual signals (prototype similarity and semantic alignment) as a gentle geometric guide rather than a hard constraint. Our ablation studies show that performance remains stable across a broad range ([0.01, 0.02]). We utilized the same default value to achieve SOTA results on Woof, Nette, and ImageNet1K without any tuning.
> We further validate robustness on diverse datasets using fixed hyperparameters. ProtoVAR outperforms baselines without any tuning.
>
> Dataset|Resolution|RDED|CaO2|ProtoVAR
> -|-|-|-|-
> CIFAR-10|32x32|37.1±0.3|39.0±1.5|44.6±0.9
> CIFAR-100|32x32|42.6±0.2|48.9±0.3|53.2±0.3
> Tiny-ImageNet|64x64|41.9±0.2|46.3±0.5|51.6±0.2
> Food-101|512x512|22.8±1.1|-|28.3±0.8
>
> **W3.Foundational Models**
>
> Pre-training dependency is a universal "cold start" challenge for generative distillation (Minimax, CaO2). However, VAR's coarse-to-fine prediction shortens the optimization path, cutting foundational training costs by >50% compared to continuous denoising in DiT:
>
> Model|Params|Epochs|A100Hrs
> -|-|-|-|
> SD2.1|~1B|-|~200K
> DiT|0.6-1B|1400+|~24.6K
> VAR-d30|~2B|350|9657
> VAR-d20|0.6B|250|3012
>
> Additionally, training ResNet50 takes mere minutes (<0.05% compute). Moreover, ProtoVAR matches/exceeds SOTA even using a randomly initialized ResNet (please see Reviewer q9Su, W2).
>
> We scale to generative methods because standard matching suffers from gradient explosion on massive datasets. VAR outperforms diffusion models with significantly lower from-scratch training costs and >70x faster generation, ensuring ProtoVAR’s compute efficiency. Our efficiency advantage over baselines is detailed in Response to Rndg (W1.2).
>
> **W4\Q2.Evaluation on ViTs**
>
> The prevailing use of CNNs (particularly ResNets) over ViTs for dataset distillation evaluation stems from architectural differences. Lacking CNNs' spatial inductive biases, ViTs are data-hungry. Training ViTs on compact datasets (e.g., 10-50 IPC) causes severe overfitting. Consequently, ResNets are adopted to evaluate the distilled datasets' information content, isolating data quality from the confounding factor of ViTs' data requirements.
>
> While CNNs serve as our primary benchmark, we already evaluated cross-architecture transfer to ViTs. As detailed in Tab.4, under the standard protocol across five subsets, ProtoVAR consistently outperforms baselines, proving our distilled knowledge effectively transfers to ViTs. Furthermore, we conducted experiments training ViTs on distilled datasets (IPC 10/50 for ImageWoof, ImageNette, and ImageNet-1K).
>
> Method|Woof10|Woof50|Nette10|Nette50|IN10|IN50
> -|-|-|-|-|-|-
> Minimax|26.5±2.3|38.2±1.7|35.8±2.2|51.4±1.9|27.3±1.8|46.9±1.3
> Ours|35.1±1.9|46.1±1.2|42.7±2.3|56.5±1.8|33.6±1.1|50.2±0.9
>
> ProtoVAR consistently outperforms Minimax, indicating that our prototype-guided generation captures richer, more robust semantic features transferable across diverse architectures.

---

> > ### Author Rebuttal · Reviewer_UJL9 · 2026-04-04
> >
> > I thank the authors for their detailed response and the additional experimental results. While I still have reservations regarding the heavy reliance on the inference speed of pre-trained models and believe the generate-and-filter approach involves a degree of structural redundancy that could be further optimized, I acknowledge that these challenges are somewhat inherent to the current state of generative dataset distillation. Considering these field-wide limitations and the performance gains demonstrated, I have decided to raise my rating to 4.

---

> > > ### Author Response · Authors · 2026-04-04
> > >
> > > We sincerely thank you for recognizing the value of our work, your understanding of the field-wide challenges, and for raising your score! We are glad that our additional experiments and responses addressed your primary concerns.
> > >
> > > We completely agree with your insightful observations regarding the structural redundancy in the "generate-and-filter" paradigm and the reliance on pre-trained models. While these remain critical bottlenecks in current generative dataset distillation, we believe that our proposed ProtoVAR takes a meaningful step forward in mitigating them compared to traditional diffusion methods, establishing a new paradigm that paves the way for fundamentally resolving these issues. For instance, exploring token-level early-exit mechanisms during VAR autoregressive generation could potentially mitigate filtering redundancy, while investigating parameter-efficient fine-tuning or lighter foundation models could further alleviate the cold-start dependency.
> > >
> > > We will further perform careful revision to explicitly include these valuable discussions in our "Limitations and Future Work" section to enhance the paper's comprehensiveness.
> > >
> > > Best regards,
> > >
> > > Authors

---

### Official Review · Reviewer_Rndg · 2026-03-11

**Soundness:** 3
**Presentation:** 3
**Significance:** 3
**Originality:** 3
**Overall Recommendation:** 5
**Confidence:** 3

**Summary:**

The authors first analyze the limitations of coreset selection methods and generative dataset distillation. Most existing generative methods rely on diffusion models, suffering from (i) indirect matching objectives and (ii) target-agnostic generation.

In this paper, the authors propose a new VAR-based framework, ProtoVAR, leveraging the complementary strengths of coreset selection and generative methods. ProtoVAR first selects class prototypes, then use prototypes to guide the VAR sampling process, and last applies target-guided filtering.

In experiments, ProtoVAR shows SoTA performance with comparable or lower computational cost than diffusion-based distillation.

**Compliance With Llm Reviewing Policy:**

Affirmed.

**Final Justification:**

Thank the authors for the very detailed and thoughtful responses. My main concerns have been largely addressed, and I appreciate their effort in clarifying the questions.

**Key Questions For Authors:**

Please see the Weaknesses (Major) section.

I am not an expert in dataset distillation, so I do not feel confident making a definitive judgment on this paper. After reviewing the comments from other reviewers and the authors’ responses, if no major issues are identified, I would be willing to support acceptance.

**Limitations:**

No. I didn't see the limitation section in the paper or the supplementary materials. The authors are encouraged to include the limitation section.

**Strengths And Weaknesses:**

## Strengths

- The motivation is clear and insightful.

- The proposed method is easy to understand and very effective.

- The experiments are comprehensive, covering different model sizes and datasets.

## Weaknesses

### Major

- I would like to know more about how the authors finally choose VAR.

    - Are other AR models or one-step diffusion generator like MeanFlow able to solve the two proposed limitations (indirect matching objectives and target-agnostic generation), as long as the sampling does not involve multi-step denoising process? Are there any advantages of VAR over those models that motivates the authors to choose VAR?

    - Compared to diffusion models, VAR introduces more tokens due to the multi-scale prediction and usually needs a large model size (like >1B) to get comparable generation quality. I’m wondering if VAR leads to higher GPU memory especially on high-resolution datasets.

    - Following the question above, the generation quality of VAR is usually worse than the SoTA diffusion models. So I’m wondering how important the generation quality is for dataset distillation. Do the authors think a model worse in generation may outperform others for the distillation task?

- I’m not an expert in dataset distillation. I’m not sure why the evaluation does not involve transformer-based models like ViT. It seems that most of experiments are based on ResNet.

- Eq 6 is not very convincing to me. The authors define the expected semantic embedding here and will use it for similarity calculation. But VAR is a discrete generative model, the expected embedding may lie between real token embeddings and therefore does not correspond to any valid token. And in sampling, such an expected embedding will not be sampled or decoded. Thus, I’m not sure if this is a reasonable way to calculate the similarity.

### Minor

- Some results of CaO2 and ProtoVAR are very close to each other. For instance, in Table 1 on ImageNet-1k, “53.0±0.2 vs 53.5±0.5”, “52.2±1.1 vs 52.7±1.0”. I’m wondering if these results are really significant.

- Figure 2, Step I, the second equation, the brackets need to be refined.

---

> ### Author Rebuttal · Authors · 2026-03-31
>
> Thank you for your detailed comments. Please find the responses below:
>
> **W1.1.Motivation for Choosing VAR**
>
> One-Step Diffusion (e.g., MeanFlow) is efficient. However, it fails for the following reasons:
> * Dataset distillation requires multi-scale guidance. We need to align the global layout with a object first, and then allow freedom for fine details. Because MeanFlow maps noise to the target in one step, there are no intermediate stages where we can apply this guidance.
> * Forcing a single-step output to match a prototype requires expensive gradient backpropagation to the input noise. Moreover, this single-step constraint causes the generation to collapse into a single point in the feature space, destroying the intra-class diversity essential for a robust dataset.
>
> While AR models provide the discrete steps, their 1D token-by-token generation (typically top-left to bottom-right) introduces critical limitations:
> * Flattening an image into a 1D sequence destroys the dense, bidirectional 2D relationships inherent in natural images. Applying global prototype guidance to early local tokens (e.g., the top-left corner) lacks global context. This blind guidance leads to localized errors that compound as generation proceeds.
> * Generating high-resolution images token-by-token is excessively slow and expensive for large-scale dataset distillation.
>
> VAR abandons the 1D top-left-to-bottom-right approach for a coarse-to-fine (resolution-by-resolution) generation, directly solving the limitations of both one-step diffusion and standard AR models:
> * VAR's parallel, hierarchical token prediction preserves 2D layouts and enables discrete multi-scale guidance. Rather than forcing exact matches, our direction-aware Logit Refinement gently boosts prototype-aligned tokens, achieving precise semantic alignment and image stability while maintaining high diversity.
> * Parallel token prediction makes VAR incredibly fast—over 70x faster than traditional diffusion transformers.
>
> **W1.2.Memory Efficiency**
>
> While VAR backbones may have higher parameter counts, ProtoVAR is significantly more memory-efficient than diffusion-based distillation in practice. This efficiency is driven by two factors:
>
> Despite processing more tokens, VAR maintains a stable VRAM footprint as resolution or model size scales. This efficiency stems from its pyramidal design, where early stages (e.g., 1 × 1 to 4 × 4) consume negligible memory. Additionally, VAR applies KV Cache compression (e.g., ScaleKV); since refiner layers focus on local details, high-resolution KV cache can be reduced by up to 90\% without quality loss.
>
> Model|Res.|BS|VRAM
> -|-|-|-
> VAR|512²|64|<10G
> Inf-DiT|512²|1|~22G
>
> Traditional diffusion distillation requires tracking massive computational graphs across timesteps for backpropagation, consuming 20-80G VRAM. In contrast, ProtoVAR uses a frozen backbone. Our Prototype-Guided Logit Refinement is a pure forward-pass algebraic adjustment requiring zero historical gradients.
>
> Method|VRAM(G)|Time(min)
> -|-|-
> Ours|8-20*|1-10*
> Minimax|23|80
> LD3M|29.4|600
> *Varies by pool size.
>
> **W1.3.Visual Expressiveness**
>
> While generative models optimize for visual realism (FID), dataset distillation prioritizes expressiveness. To achieve low FID, diffusion models like DiT often produce redundant, average samples. Under strict budgets (10/50 IPC), this lack of diversity easily leads to overfitting. Conversely, images that highlight core, diverse class features even if less realistic provide much stronger and more accurate training gradients.
>
> VAR builds images scale-by-scale. Each step maintains clear structural and semantic integrity, allowing injection of class features without noise interference. This yields "information-dense," effective training images.
> Despite prioritizing expressiveness, ProtoVAR remains visually competitive, consistently outperforming diffusion baselines in FID evaluated on the ImageNet-1K training set. ProtoVAR preserves visual fidelity while significantly enhancing semantic discriminability
>
> Method|FID(10IPC)|FID(50IPC)
> -|-|-
> Minimax|18.6|15.0
> VAR|17.2|12.9
> Ours|17.0|13.1
>
> **W2.Evaluation on ViTs**
>
> Please refer to W4 of the response for Reviewer UJL9.
>
> **W3. Analysis of Eq. 6 \& 7**
>
> Please refer to W1 of the response for Reviewer 77gS.
>
> **W4. Performance Gains**
>
> In the highly constrained setting of compressing 1,000 classes into 10 IPC, an absolute gain of 0.5% on ImageNet-1K is a significant milestone.
> * Our gains are consistent across evaluation architectures (+0.7%, +0.5%, and +0.5% on ResNet-18, ResNet-50, and ResNet-101).
> * Unlike CaO2, which requires costly latent optimization (e.g., 20 minutes on ImageWoof IPC 10), ProtoVAR directly guides generation via a single forward pass (1-10 minutes). ProtoVAR advances SOTA not just in accuracy, but by eliminating the fundamental speed bottlenecks of diffusion distillation.
>
> **W5. Brackets**
>
> Thank you for the careful reading. We have correctly scaled the brackets.

---

> > ### Author Rebuttal · Reviewer_Rndg · 2026-04-01
> >
> > Thank the authors for the detailed experiments and the thoughtful responses. I also read the comments from the other reviewers and the corresponding replies. The authors have provided thorough experimental evidence that addresses my concerns. I will raise my score and support the acceptance of this paper.

---

> > > ### Author Response · Authors · 2026-04-01
> > >
> > > We sincerely thank the reviewer for recognizing the value of our work and supporting its acceptance! We are glad that our additional experiments and responses addressed your concerns. We will further perform careful revision to address the comments and enhance the paper quality.
> > >
> > > Best regards,
> > >
> > > Authors

---

### Official Review · Reviewer_q9Su · 2026-03-11

**Soundness:** 3
**Presentation:** 3
**Significance:** 2
**Originality:** 3
**Overall Recommendation:** 4
**Confidence:** 4

**Summary:**

This paper introduces ProtoVAR for dataset distillation using visual autoregressive modeling. The approach extracts multi scale class prototypes via coreset selection to address the semantic ambiguity of noise dominated diffusion states. In the coarse to fine generation phase an expected semantic embedding from predicted logits estimates the trajectory. A direction aware logit bias based on cosine similarity with reference prototypes constrains token sampling. A pretrained teacher classifier subsequently selects high confidence synthetic samples to construct the dataset.

**Compliance With Llm Reviewing Policy:**

Affirmed.

**Final Justification:**

Thank you to the authors for the thorough and constructive rebuttal. The additional experiments and detailed clarifications have successfully addressed my main concerns.

**Key Questions For Authors:**

Table 2 shows a performance decrease without target guided filtering. The authors should compare the ProtoVAR method without this filtering against baselines including Minimax and D^3HR under identical settings. A comparison is necessary to evaluate the VAR generation mechanism against diffusion models without the influence of the teacher model.

If the baselines achieve comparable or better results after applying your filtering module, the core contribution of using VAR over diffusion models would be significantly weakened.

What happens to the distillation performance if the pre-trained ResNet-50 (used for extracting multi-scale prototype descriptors) is replaced by a randomly initialized network or a significantly smaller architecture?

**Limitations:**

yes

**Strengths And Weaknesses:**

**Strengths**

The next scale parallel prediction in VAR decreases the computational costs of sequential denoising. Prototype guidance functions within the forward pass and avoids secondary optimization or fine tuning. Proposition 1 demonstrates that softmax normalization nullifies uniform prototype biasing which justifies the direction aware token perturbation. Experiments on ImageNet subsets and ImageNet 1K across ResNet EfficientNet and MobileNet validate the cross architecture generalization of the synthesized datasets.

**Weaknesses**
1. The proposed method applies a pretrained teacher classifier for target guided filtering. Generative baselines such as Minimax and D^4M do not use external classifiers for selection. Evaluating the method against these baselines creates an unfair comparison due to the asymmetric external knowledge. The observed accuracy gains likely result from the teacher classifier rather than the prototype guided VAR generation.

2. The method uses pretrained ResNet 50 for prototype extraction and ResNet 18 for filtering. The manuscript lacks ablation studies evaluating the capacity of these models. The authors should clarify how the framework performs when using weaker or randomly initialized feature extractors. This missing evaluation reduces the methodological completeness.

---

> ### Author Rebuttal · Authors · 2026-03-31
>
> Thank you for your detailed comments. Please find the responses below:
>
> **W1.Fair Comparison on Filtering**
>
> We agree that a teacher classifier might seem unfair. However, target-guided filtering is not just an extra step. It is a practical capability unlocked by VAR's efficiency. Filtering requires generating a large candidate pool. For ImageNet1K at IPC 50, this means generating ~250k high-resolution images. Forcing  Diffusion like DiT to generate a 5x pool via sequential denoising incurs massive GPU costs, defeating the efficiency purpose of dataset distillation. Conversely, VAR's parallel prediction is ~70x faster than DiT. This efficiency enables fast, cost-effective sampling, making large-scale filtering viable for VAR but impractical for diffusion models.
>
> We evaluated ProtoVAR without the teacher classifier: (a) w/o pool (generates target IPC directly), and (b) w/ pool+rand (generates 5x pool, selects randomly).
>
> Method|Woof10|Woof50|Nette10|Nette50
> -|-|-|-|-
> Minimax|40.1±1.0|67.0±1.8|61.4±0.7|84.1±0.2
> $D^4M$|37.5±1.8|65.7±1.7|59.3±1.4|82.5±1.4
> $D^3HR$|41.1±0.8|70.3±0.9|62.8±0.7|84.6±1.2
> ProtoVAR(w/o)|43.2±0.7|69.5±1.1|65.3±0.8|82.1±1.4
> ProtoVAR(rand)|45.1±1.1|70.6±1.3|66.3±1.3|84.2±1.5
>
> Even without filtering, ProtoVAR outperforms methods at IPC 10 and is highly competitive at IPC 50. The slightly lower performance at IPC 50 compared to $D^3HR$ is expected: prototype guidance ensures representativeness, but generating a larger pool is needed to push the upper limits of diversity. ProtoVAR achieves strong results in a fraction of the time required by diffusion models. This shows that even without filtering, ProtoVAR is highly cost-effective and drastically faster than baselines:
>
> Method|Time/Img
> -|-
> ProtoVAR|~0.08s
> DiT-XL/2|~0.8s
> DDIM|~1.2s
>
> To ensure fairness, we applied target filter to baselines:
>
> Method(+Filter)|Woof10|Woof50|Nette10|Nette50
> -|-|-|-|-
> Minimax|41.7±1.5(↑1.6)|68.9±2.1(↑1.9)|62.9±0.9(↑1.5)|84.9±1.3(↑0.8)
> $D^4M$|39.6±1.7(↑2.1)|67.0±1.9(↑1.3)|60.2±1.8(↑0.9)|83.6±1.5(↑1.1)
> $D^3HR$|44.3±1.1(↑3.2)|71.6±1.2(↑1.5)|64.2±1.0(↑1.4)|85.6±1.3(↑1.0)
> ProtoVAR|47.2±1.3(↑4.0)|72.0±0.7(↑2.5)|68.0±0.9(↑2.7)|85.8±0.8(↑3.7)
>
> While filtering improves baselines, ProtoVAR still outperforms them. Generating this large pool drops the efficiency of diffusion models. ProtoVAR's boost is synergistic: while prototype guidance anchors representativeness, VAR explores stylistic diversity. This rich diversity produces some samples with variations that are harder to recognize. The filter simply remove these ambiguous cases, keeping diverse, accurate samples.
>
> **W2.Ablation on Model Capacity**
>
> We agree that evaluating model capacity strengthens framework completeness. Accordingly, we tested the prototype extractor and teacher filter (random, weaker, default, larger models).
>
> We first fixed teacher filter and varied the backbone used for feature extractor:
>
> Feature Extractor|Woof10|Woof50|Nette10|Nette50
> -|-|-|-|-
> Rand(ResNet50)|46.3±1.2|71.3±1.5|66.9±1.4|85.2±1.6
> ConvNet|46.5±1.1|71.6±1.3|67.8±1.2|85.2±1.4
> MobileNetV2|46.9±0.9|71.8±1.1|68.1±1.0|85.5±1.1
> ResNet50(Ours)|47.2±1.3|72.0±0.7|68.0±0.9|85.8±0.8
> ResNet101|47.4±1.1|71.9±0.8|68.2±1.1|85.6±0.9
>
> Performance remains highly stable across different model. Notably, a randomly initialized ResNet50 causes a minor accuracy drop. This aligns with DM (WACV 2023) findings that random networks can effectively capture data distributions. Since ProtoVAR uses the extractor solely to compute simple cosine similarities, random networks provide sufficient guidance. Scaling up from ConvNet to ResNet101 yields marginal gains. We default to ResNet50 for its standardized feature space, but these results confirm ProtoVAR does not rely on a high-capacity extractor.
>
> Next, we fixed the extractor and varied the teacher network.
>
> Teacher Filter|Woof10|Woof50|Nette10|Nette50
> -|-|-|-|-
> Rand(ResNet18)|45.1±1.1|70.6±1.3|66.3±1.3|84.2±1.5
> ConvNet|45.9±0.9|71.2±1.0|66.8±1.0|84.7±1.2
> MobileNetV2|46.3±0.8|71.4±0.9|67.1±0.9|85.2±1.0
> ResNet18(Ours)|47.2±1.3|72.0±0.7|68.0±0.9|85.8±0.8
> ResNet50|47.3±1.1|72.2±0.8|68.0±1.0|85.9±0.7
>
> Random initialization reduces filtering to random sampling to form the pool. While performance drops, the results still match or exceed current SOTA methods. Introducing lightweight pre-trained models (e.g., ConvNet, MobileNet) restores performance, as they effectively identify high-quality, in-distribution samples. Upgrading to ResNet50 provides marginal gains. This confirms that ResNet18 offers the optimal trade-off between filtering accuracy and computational efficiency.
>
> These results confirm that ProtoVAR does not require massive models. While pre-trained networks provide a performance boost, randomly initialized models deliver gains that exceed current methods. Therefore, our selection of ResNet50 for extraction and ResNet18 for filtering provides the optimal trade-off between distillation performance and computational accessibility.

---

> > ### Author Rebuttal · Reviewer_q9Su · 2026-04-03
> >
> > The author has provided comprehensive experimental evidence, effectively addressing my concerns

---

> > > ### Author Response · Authors · 2026-04-03
> > >
> > > We sincerely appreciate your thoughtful review and consideration of our rebuttal. We are pleased that our clarifications addressed your concerns, and we are grateful for your positive assessment. We will further perform careful revision to enhance the paper quality.
> > >
> > > Best regards,
> > >
> > > Authors

---

### Official Review · Reviewer_77gS · 2026-03-12

**Soundness:** 2
**Presentation:** 2
**Significance:** 2
**Originality:** 2
**Overall Recommendation:** 4
**Confidence:** 2

**Summary:**

This paper proposes ProtoVAR, a dataset distillation method based on a prototype-guided visual autoregressive (VAR) framework. The method uses a direction-aware logit refinement strategy to push VAR sampling toward multi-scale, class-specific prototypes through an expected-embedding surrogate. It also has a target-guided cleaning stage with a teacher model, then a coreset-style selection step, to get compact and useful distilled datasets. Experiments on ImageWoof, ImageNette, and ImageNet-1K show that the method keeps beating diffusion-based distillation baselines. It is also more training-efficient, since it does not need extra generator fine-tuning.

**Compliance With Llm Reviewing Policy:**

Affirmed.

**Final Justification:**

Most of my concerns are addressed. I will raise my score to 4.

**Key Questions For Authors:**

See weaknesses.

**Limitations:**

yes

**Strengths And Weaknesses:**

- Strengths
	- The paper presents a fairly principled and lightweight way to guide autoregressive sampling with expected embeddings and prototype similarity. Compared with diffusion-based approaches, this design avoids some of the instability caused by noise-corrupted diffusion latents.
	- The method is tested under multiple IPC settings on ImageWoof, ImageNette, and ImageNet-1K. The experiments also include several architectures, such as ResNet-18, ResNet-50, and ResNet-101.
- Weaknesses
	- The expected embedding in Eq. (6) is insufficiently justified. Averaging token embeddings under the predicted distribution may not yield a semantically faithful representation of the actual discrete sampling trajectory. This concern is made worse by the tentative decoding and prototype similarity computation in Eq. (7), since passing a soft averaged representation through a decoder designed for discrete codes may introduce additional bias.
	- The target-guided filtering part is still not explained clearly enough. The choice of teacher model, and its relation to the evaluation backbones, are still a bit unclear.

---

> ### Author Rebuttal · Authors · 2026-03-31
>
> Thank you for your detailed comments. Please find the responses below:
>
> **W1. Analysis of Eq. 6 \& 7**
>
> To quantify the gap between the true expected objective of discrete sampling and continuous surrogate in Eq. 6, we analyze the objective $J(h)$ using a second-order Taylor expansion. Let $\Pi$ be the predicted probability distribution. The true expected objective of discrete sampling trajectory is $\mathbb{E} _ {v \sim \Pi}[J(E_v)]$. We uses the expected embedding $\bar{h} = \mathbb{E}_{v \sim \Pi}[E_v]$ to compute a surrogate objective $J(\bar{h})$. Expanding $J(E_v)$ around $\bar{h}$ and taking the expectation yields:
>
> $$\mathbb{E}_{v \sim \Pi}[J(E _ {v})] = J(\bar{h}) + \nabla J(\bar{h})^\top \mathbb{E} _ {v \sim \Pi}[E _ {v} - \bar{h}] + \frac{1}{2} \mathbb{E} _ {v \sim \Pi}\left[ (E _ {v} - \bar{h})^\top \nabla^2 J(\xi _ {v}) (E _ {v} - \bar{h}) \right]$$
>
> Since $\mathbb{E}_{v \sim \Pi}[E_v - \bar{h}] = 0$, the first-order gradient term vanishes. Assuming a bounded Hessian (Lipschitz constant $L$), the surrogate error is bounded by the second-order variance (covariance $\Sigma$): $|\mathbb{E} _ {v \sim \Pi}[J(E_v)] - J(\bar{h})| \leq \frac{L}{2} \text{Tr}(\Sigma)$. In a well-trained VAR, probability mass collapses toward optimal tokens during autoregressive generation. As this logit concentration occurs, $\text{Tr}(\Sigma) \to 0$, shrinking the error bound to 0 and proving $\bar{h}$ tightly tracks the true discrete trajectory.
>
> Regarding potential bias in tentative decoding (Eq. 7), the VQ-VAE decoder $D(\cdot)$ is Lipschitz continuous, meaning visual deformation from soft decoding is strictly bounded ($\|D(\bar{h}) - D(E_v)\| \le L_D \|\bar{h} - E_v\|$). Furthermore, the tentatively decoded image immediately passes through a deep CNN feature extractor. The CNN inherently acts as a semantic low-pass filter, robust to micro-level pixel perturbations and focusing entirely on global macro-semantics, effectively washing out minor decoding artifacts.
>
> Architecturally, tentative decoding merely serves as an informational probe. If high uncertainty yields a blurry image, prototype similarity ($sim_b^{(s)}$) drops to zero, naturally fading out the guidance. The actual intervention remains a minimal first-order perturbation ($\lambda \ll 1$) applied to the logits. The final generation trajectory never physically leaves the discrete network, guaranteeing perfectly sharp, artifact-free images.
>
> We calculated the cosine similarity (in the ResNet50 feature space) between images generated via hard sampling and soft decoding. It consistently scores between **0.94** and **0.98** in mid-to-late stages, proving that the soft decoding trajectory is macro-semantically isomorphic to the discrete trajectory. Additionally, We tracked the Top-1 probability of the model's predictions over the generation steps:
>
> Step|0|1|2|3|4|5|6|7|8|9
> -|-|-|-|-|-|-|-|-|-|-
> Top-1|0.92|0.62|0.79|0.82|0.85|0.87|0.90|0.95|0.97|0.99
>
> The data confirms a probability collapse, verifying that the variance $\text{Tr}(\Sigma)$ and thus our error bound quickly converges to a negligible constant.
>
> Finally, our continuous approximation design aligns perfectly with recent foundational literature. Works such as Soft-Di[M]O (ICLR 2026) and SoftVQ-VAE (CVPR 2025) mathematically validate that continuous topological averaging and expected embeddings seamlessly integrate with VQ decoders, yielding higher quality than strictly discrete lookups. Concurrently, AREdit (CVPR 2025) and EAR (ICML 2025) demonstrate that relying on hard tokens during guidance causes significant information loss, advocating for the use of complete probability distributions, precisely the mechanism our method successfully employs.
>
> **W2.Clarification of the Target-Guided Filtering**
>
> Generative models are inherently target-agnostic. We introduce Target-Guided Filtering as a lightweight quality-control step. We default to ResNet18 as the teacher for two reasons:
> * Efficiency: Filtering requires scoring many samples; ResNet18 is fast and lightweight.
> * Reliability: It provides stable decision boundaries to verify core class semantics, acting as a reliable proxy for sample quality.
>
> Our method decouples generation from evaluation architecture, while the teacher is used only for confidence-based filtering. The distilled dataset is inherently architecture-agnostic. As shown in Tab.4 and Fig.9, when ResNet18 is used during filtering, it achieves strong transfer performance to other backbones (e.g., EfficientNet, ViT). To further prove that downstream performance depends on the teacher's capacity rather than structural similarity to the evaluator, we tested additional pairs on Woof10:
>
> Teacher Filter|ResNet18|ConvNet|MobileNet|ViT
> -|-|-|-|-
> ConvNet|45.9±1.2|39.4±1.3|45.1±1.2|44.9±1.1
> MobileNet|46.3±1.3|40.2±1.2|45.6±1.0|45.5±1.0
> ResNet18(Ours)|47.2±1.3|40.6±1.1|46.3±0.9|45.1±0.8
>
> Please refer to W2 of the response for q9Su, a simple model suffices independently of the downstream architecture.

---

> > ### Author Rebuttal · Reviewer_77gS · 2026-04-02
> >
> > Thanks for your response. Most of my concerns are addressed. I will raise my score to 4.

---

> > > ### Author Response · Authors · 2026-04-02
> > >
> > > We sincerely appreciate your positive feedback and the updated score. We are very glad that our rebuttal adequately addressed your concerns. We will carefully revise the manuscript again to move some important results to the main text.
> > >
> > > Best regards,
> > >
> > > Authors

---

### Decision · Program_Chairs · 2026-04-30

**Decision:**

Accept (regular)

**Comment:**

This paper introduces ProtoVAR, a prototype-guided dataset distillation framework built on visual autoregressive (VAR) modeling that directly addresses key limitations of diffusion-based distillation. By integrating multi-scale class prototypes, direction-aware logit refinement, and efficient pool-based selection, the method produces representative and diverse distilled datasets with state-of-the-art performance and significantly improved efficiency. Following the rebuttal, the paper received unanimous support from all reviewers, leading to an Accept decision.